# Dynamical phases in a "multifractal" Rosenzweig-Porter model

Ivan M. Khaymovich[1,2]⋆ and Vladimir E. Kravtsov[3,4]

**1** Max-Planck-Institut für Physik komplexer Systeme,
Nöthnitzer Straße 38, 01187-Dresden, Germany
**2** Institute for Physics of Microstructures, Russian Academy of Sciences,
603950 Nizhny Novgorod, GSP-105, Russia
**3** Abdus Salam International Center for Theoretical Physics,
Strada Costiera 11, 34151 Trieste, Italy
**4** L. D. Landau Institute for Theoretical Physics, Chernogolovka, Russia

⋆ ivan.khaymovich@gmail.com

## Abstract

We consider the static and the dynamical phases in a Rosenzweig-Porter (RP) random matrix ensemble with a distribution of off-diagonal matrix elements of the form of the large-deviation ansatz. We present a general theory of survival probability in such a random-matrix model and show that the *averaged* survival probability may decay with time as a simple exponent, as a stretch-exponent and as a power-law or slower. Correspondingly, we identify the exponential, the stretch-exponential and the frozen-dynamics phases. As an example, we consider the mapping of the Anderson localization model on Random Regular Graph onto the RP model and find exact values of the stretch-exponent $\kappa$ in the thermodynamic limit. As another example we consider the logarithmically-normal RP random matrix ensemble and find analytically its phase diagram and the exponent $\kappa$. Our theory allows to describe analytically the finite-size multifractality and to compute the critical length with the exponent $\nu_{MF} = 1$ associated with it.

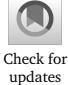
# 1 Introduction

Random matrix theory (RMT) has tremendous range of applications in physics spanning from the spectra of heavy nuclei to physics of glassy matter. Specifically, it was a basis for mesoscopic physics with a very successful applications to the description of ergodic phases in disordered condensed matter systems both on the single-particle level, as in diffusive metals, and in the many-body problem of thermalization in generic isolated quantum systems. In the latter problem, the paradigm of RMT has been further extended to the *eigenstate thermalization hypothesis* (ETH) [1–3] claiming the thermalization in quantum many-body systems isolated from a thermal bath. Indeed, ETH uses the fact [1] that close-in-energy eigenstates of a chaotic Hamiltonian hybridize with essentially random phases under a small perturbation and thus can be represented by eigenvectors of the Gaussian random matrix ensembles (GRME) or by the so-called random pure states. In both descriptions the states have the (nearly) i.i.d. Gaussian-distributed wave function coefficients and they are used not only for the description of the equipartition over degrees of freedom, or ETH, but also provide the ground for thermalization based on the scrambling [4] and entanglement [5] of these degrees of freedom. In all these fields, RMT provides an excellent statistical description of the problems highlighting the universal properties shared by all kinds of chaotic systems.

The phenomenon of Anderson localization was the simplest example of the failure of the classical RMT of Wigner and Dyson (WD RMT) on the single-particle level. In order to describe Anderson localization that happens in a certain (laboratory) frame one has to introduce the *basis preference* in RMT. The simplest modification is the Rosenzweig-Porter RMT [6] which is nothing but the WD RMT with the special, stronger fluctuating diagonal matrix elements setting the basis preference.

During the recent decade the phenomenon of the many-body localization (MBL) was discovered [7, 8] and quickly developed into a whole field of the thermalization breakdown in

generic isolated quantum many-body systems under the sufficiently strong quenched disorder [9]. At such conditions ETH is strongly violated raising a question about a model which is as simple as WD RMT but capable of describing the generic features of the MBL and pre-MBL states.

One may start by the Fock/Hilbert space formulation of the MBL problem for models with local interactions in the coordinate basis. A crucial simplification on this way was suggested in the seminal paper [7] where it was assumed that the Fock/Hilbert space of such systems form a *hierarchical graph* like the Cayley tree (CT) or the Random Regular Graph (RRG) [10] and the MBL state is the Anderson localized state in the Hilbert space. This boosted the field of Anderson localization on hierarchical graphs with the local tree-like structure suggested [11–14] as a toy model of MBL in the Hilbert space of such many-body models.

However, in a number of recent works [15–19] such a picture has been put in doubt. Namely, due to the locality of the Hamiltonian in the coordinate (often one-dimensional) space, absence of transport in the coordinate space does not mean localization in the Hilbert space. A simple model of the small fraction $0 < f < 1$ of frozen spins [15] in a 1D $s$-spin chain of length $L$ with nearest neighbor interaction shows that the wave function support set in the Hilbert space has a volume $(2s + 1)^{(1-f)L} = N^{(1-f)}$, where $N = (2s + 1)^L$ is the total volume of the Hilbert space, i.e. it has a Hausdorff dimension $0 < D = 1 - f < 1$ and thus is a fractal. At the same time frozen spins block the spin-excitation transport completely. A little more elaborate model that involves the localized *one-particle* states *with finite localization radius* and the Slatter-determinant many-body states shows that the inverse-participation ratio of such many-body states corresponds to the fractal dimension $0 < D_q < 1$ even without interaction [18]. The role of interaction in this case is played by the entanglement and Pauli Principle encoded in the structure of Slatter determinants. Small repulsive interaction does the same job for vanishing localization radius. Thus multifractality is intrinsic to many-body interacting systems. Such states are neither localized nor fully ergodic. They occupy a sub-extensive part of the whole accessible Hilbert space and form an eigenstate multifractality in the Hilbert space in a whole MBL phase.

The above mentioned Anderson models on the hierarchical graphs have been suggested [11–13] as the toy models that support such a robust non-ergodic extended (NEE) phase, but the existence of such a phase in the thermodynamic limit has been disputed during recent years [13, 14, 20–24].

In contrast, in the random-matrix theory such systems with the NEE phase have been recently found. It appears that the simplest basis non-invariant Rosenzweig-Porter ensemble [6] supports the NEE phase if the variance of the i.i.d. diagonal matrix elements is parametrically larger $\langle H_{nn}^2 \rangle = N^\gamma \langle |H_{n \neq m}|^2 \rangle$ than that of the off-diagonal ones drawn from the GRME, where $N$ is the matrix size. It was discovered in Ref. [25] with the further rigorous mathematical proof in Ref. [26] that the eigenvectors of the RP model with $\gamma$ in the interval $1 < \gamma < 2$ have a fractal support set with the Hausdorff dimension $D = 2 - \gamma$. This example has been intensively studied [27–32] over the past few years.

It is conceptually important that the non-trivial NEE phase in the RP model arises only if the statistics of matrix elements depends on the matrix size in a proper way described above. We will show in this paper that the models with *short-range, size-independent* couplings, such as models with the nearest-neighbor coupling on RRG, can be mapped onto the *infinite-range* RP models with the distribution of matrix elements *depending on the matrix size*. This dependence is the price for a relative ease to treat the infinite-range matrix models. The first demonstration of such type of mapping was done by Efetov [33] when he reduced the $d$-dimensional Anderson model with a sparse matrix Hamiltonian to the infinite-range GRME for frequencies $\omega$ smaller than the Thouless energy by means of the supersymmetric non-linear sigma-model. In this case rescaling of all the matrix elements by any $N$-depending factor does not change

the properties of GRME. However, to make the spectral bandwidth finite $E_{BW} = 1$ one has to make $N$-dependent the Gaussian statistics of matrix elements in GRME.

Reducing the Anderson models on the hierarchical graphs to a certain kind of the RP models is the next non-trivial step. However, the Gaussian RP (GRP) model considered in Ref. [25] is rather oversimplified to be a good candidate for this goal.

Because of the infinite connectivity and self-averaging of the matrix element fluctuations the exact solution for this model can be written in the standard Breit-Wigner form [27, 30, 31]

$$|\psi_\alpha(n)|^2 \sim \frac{C}{(E_\alpha - \varepsilon_n)^2 + \Gamma^2} \ , \tag{1}$$

with the certain normalization constant $C$ and the level broadening $\Gamma$ given by the escape rate from the Fermi Golden rule

$$\Gamma = \frac{2\pi}{\hbar} \rho(E) \sum_n |H_{mn}|^2 \ . \tag{2}$$

The self-averaging of the above sum in the thermodynamic limit (confirmed by the cavity method calculations [27, 31]) immediately supports the Breit-Wigner approximation and the diffusive transport [29, 32] in such a model.

In more realistic local many-body systems, the corresponding matrix in the reference Hilbert space basis (e.g. in the "computational" basis of strings, or "configurations" of up/down spins in the spin-1/2 spin chain) is sparse and the *effective connection* of far-away configurations (strings) is determined by the presence of a long series of quantum transitions and anomalously strong resonances. These effective long-range hopping amplitudes are in general correlated and fat-tailed distributed [17, 18, 34].

The main idea [17, 35] of a random matrix representation of the many-body problem in the Hilbert space is to replace the sparse matrix of the many-body Hamiltonian by the full matrix of the RP type with off-diagonal elements representing the *effective* connections. This is equivalent to reduction of a quantum disordered problem on a hierarchical graph to that on a complete graph with a proper statistics of matrix elements. This mapping is essentially a reduction of an Anderson localization problem on an *infinite-dimensional* lattice to that on a *zero-dimensional* one, in which the underlying hierarchical graph structure disappears but remains encoded in the statistics of matrix elements.

An important simplification comes from the "infinite-temperature" condition. Indeed, the statistics of highly-excited states is supposed to be the same for a broad band of such states which occupy a certain part of the Hilbert space. In this fraction of the Hilbert space there is an approximate statistical homogeneity over the corresponding configurations, very much like the statistical homogeneity over sites on the Random Regular Graph. This homogeneity is a crucial point for the application of the RP-type random matrix theory with statistical homogeneity over matrix indices to realistic many-body systems. Apparently, a reduction to RP random matrix theory is not an exact mapping. Some information is inevitably lost after the "folding" of the hierarchical Hilbert space into a "zero-dimensional" RP model. For instance it is no longer possible to define the diffusion over configurations in the Hilbert space but it is possible to study the time-dependence of survival probability in a given configuration which contains the same information as the diffusion in the (Hilbert) space. It is also possible to study the Hilbert space multifractality via distribution function of the many-body wave-function coefficients.

The main difference between the Gaussian RP model and those effective RP models that arise in the localization problem on hierarchical graphs [35] and in realistic many-body systems [17] is that the distribution of off-diagonal matrix elements in the latter is *heavily tailed*. One consequence of that is that the wave functions in the NEE phase of such RP models are genuinely *multi*fractal and not mono-fractal, as in Gaussian RP model [25].

In this paper we concentrate on another important consequence of the broad distribution of off-diagonal matrix elements in such RP models. Namely, we show that the survival probability in a broad region of parameters shows the stretch-exponential behavior rather than the simple exponential one present in the Gaussian RP model [32]. Since the diffusion on a hierarchical graph (like in a well-known example of a Cayley tree) results in the simple exponential decay of survival probability, the stretch-exponential behavior should be associated with the *sub-diffusion*. This observation is important because even on the extended side of the MBL transition in some many-body systems the diffusive character of transport is not completely confirmed by several numerical calculations [36–44]. This leaves a room for the subdiffusive dynamics, probably by forming the corresponding phase with the anomalously slow (glassy) transport properties due to some bottleneck originating from the Griffiths-type physics (see reviews [45, 46] and references therein).

In this paper we demonstrate that in some important and quite wide class of RP models with a heavily-tailed, "multifractal" distribution of hopping matrix elements such phases with a stretch-exponential dynamics of survival probability are *generically* present. We derive analytically the lines of phase transitions between the exponential (E) and stretch-exponential (SE) dynamics, as well as the expression for the stretch-exponent $\kappa$ for a generic "multifractal" RP model.

## 1.1 The model

We consider a RP model with a "large-deviation", or "multifractal" distribution $\mathcal{F}(H_{mn})$ of i.i.d. off-diagonal (hopping) matrix elements of the form:

$$\mathcal{F}\left(g = -\frac{\ln |H_{nm}|^2}{\ln N}\right) = C \, N^{F(g)}, \quad |H_{nm}| = N^{-g/2} , \tag{3}$$

with the typical coupling $H_{typ} \sim N^{-\gamma/2}$, $\gamma > 0$ which is polynomially small in the dimension $N$ of the Hilbert space of the system.

Below we show (see also Ref. [35]) that it is exactly the form that emerges in the effective RP model for the Anderson localization model on RRG. In a generic many-body problem, the non-trivial $N$-dependence in Eq. (3) can be understood from the following observation: when one forms the effective long-range hopping in the Hilbert space the transition amplitude is typically determined by the product of individual local hops and thus decays exponentially $H_{mn} \sim e^{-Ar_{mn}}$ with the Hamming distance $r_{mn}$ between configurations $m$ and $n$, while the number of available configurations $n$ to hop from a certain configuration $m$ grows exponentially with $r$, resulting in $\mathcal{F}(H_{mn}) \sim e^{Br_{mn}}$. As the most frequent inter-configuration distances $r_{\max}$ are of the order of the Hilbert space diameter, $r \simeq r_{\max} \sim \ln N$, this leads to $H_{mn} \sim N^{-g/2}$ and $\mathcal{F}(H_{mn}) \sim N^B$. Finally, taking into account large deviations of highly fluctuating matrix elements $H_{mn}$ from the typical ones, one has in general to write $F(g)$ instead of $F = B$. The properties of the function $F(g)$ are determined by the underlying local structure of the sparse-matrix Hamiltonian in the Hilbert space and may be subject to certain conditions and symmetries.

Note that the diagonal matrix elements $\varepsilon_n = H_{nn}$ in the MF-RP model are identical to on-site energies in the underlying short-range model. In what follows we consider the box-shaped distribution of on-site energies:

$$\mathcal{P}_{\text{box}}(\varepsilon) = \frac{1}{W} \, \theta(W/2 - |\varepsilon|) . \tag{4}$$

In contrast to the distribution of the off-diagonal matrix elements, Eq. (3), the distribution Eq. (4) is independent of the matrix size $N$ and *is not tailed*.

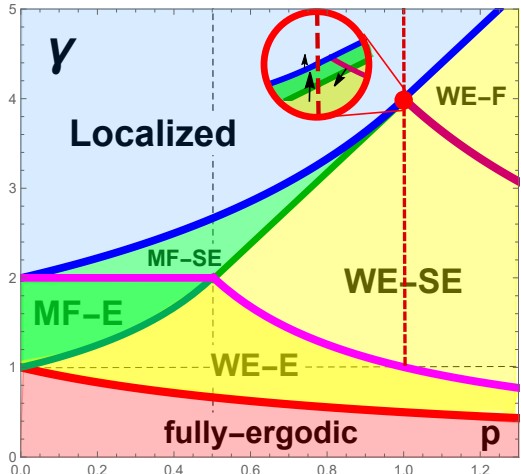

Figure 1: (Color online) **Phase diagram for the log-normal RP (LN-RP) ensemble**, $F(g) = -(g - \gamma)^2/(4p\gamma)$, in the plane of effective disorder $\gamma$ and the symmetry parameter $p$. The phase transition lines between the Localized (L), Multifractal (MF), Weakly-Ergodic (WE) and Fully-Ergodic (FE) phases are shown together with the lines separating the dynamical phases: Frozen-dynamics (F), Stretch-Exponential (SE) and Exponential (E). Albeit RP ensemble corresponding to RRG is *not exactly* log-normal, it has the same phase diagram as LN-RP with $p = 1$ and $\gamma = 2\lambda_{typ}/\ln K \geq 1$. The corresponding path in the parameter space respecting the symmetry Eq. (6) is shown by a red dashed line. As the Lyapunov exponent on a Cayley tree $\lambda_{typ} \to (1/2)\ln K$ in the clean limit, the part of the phase diagram $\gamma < 1$ is not relevant for RRG. Thus in the thermodynamic limit on RRG there is only the localized and the weakly-ergodic phase with a stretch-exponential dynamics. **In the inset:** finite-size phases in the vicinity of the tri-critical point. Like on RRG, a finite-size multifractality (FSMF) emerges near the localization transition. Arrows show the direction and "speed" of evolution as the system size $N$ increases.

Like in the previously considered Log-normal (LN) [35,47] and Levy [30,48] extensions of the RP model (which are the particular cases of $F(g)$ being quadratic or linear function of $g$), the generic MF-RP has a rich phase diagram with the robust genuinely-multifractal phase (see Fig. 1). This phase diagram can be easily understood in terms of the Breit-Wigner formula (1) (see, e.g., [31,47,49]) where the broadening $\Gamma$ is determined self-consistently via the cavity method (see, e.g., [48]).

## 1.2 Dynamical phases

We focus on the *dynamical* properties of the generic MF-RP model in order to reveal anomalously slow transport and possible subdiffusion – diffusion phase transition. Our analytical approach is based on the Wigner-Weisskopf approximation [30] which allows us to demonstrate the breakdown of the Fermi Golden rule approximation (2) and explicitly calculate the survival/return probability $R(t)$ for a quantum system to return back to the initial configuration. The latter is shown to decay either *exponentially* (corresponding to the Fermi Golden rule and the diffusive transport) or *stretch-exponentially* corresponding to the sub-diffusion (similar to those in RRG [23,24]). This *stretch-exponential* (SE) decay with the exponent $0 < \kappa < 1$

$$R(t) \propto \exp[-(E_0 t)^\kappa], \qquad (5)$$

persisting in the extensive time interval $1 \lesssim (E_0 \, t)^\kappa \lesssim \ln N$, is an emergent property of a generic MF-RP model. It may be present both in a non-ergodic (multifractal) and in a weakly-ergodic phases [1] The only difference in the SE decay in these two phases is in the scaling with $N$ of the characteristic energy (or time) scale $E_0$: in the multifractal phase this scale $E_0/E_{BW}$ (measured in units of the total spectral band-width $E_{BW}$) decays to zero in the thermodynamic limit, while in the weakly-ergodic phase this ratio $E_0/E_{BW} \sim O(1)$ remains finite.

The spreading of the stretch-exponential decay of $R(t)$ over the extensive time interval allows one to define (in the thermodynamic limit $N \to \infty$) a set of *dynamical phases* characterized by the exponent $\kappa$. In a general case we identify the following phases:

**(i)** the "frozen-dynamics phase" (F), $\kappa = 0$,

**(ii)** the "stretch-exponential phase" (SE), $0 < \kappa < 1$,

**(iii)** the "exponential phase" (E), $\kappa = 1$.

Note that the F phase does not exclude a polynomial in time relaxation of the initial configuration and thus it is not necessarily coinciding with the localized phase.

Furthermore, we derive explicit conditions for each of the phase to occur in terms of the function $F(g)$. For the particular example of the LN-RP model, $F(g) = -(g - \gamma)^2/(4p\gamma)$, we show that the transitions between the above dynamical phases do not coincide with the ones established in Ref. [47] on the basis of analysis of *ergodicity* and its violation. We show that both SE/E and F/SE transitions may occur in the multifractal phase as well as in the weakly-ergodic one (see Fig. 1) and thus the above classification of dynamical phases reflects a property of the eigenfunction statistics which is *complementary* to ergodicity and multifractality.

An important particular case is the LN-RP model with the symmetry $p = \langle \delta g^2 \rangle/(2\langle g \rangle) = 1$ which is a particular case of a general symmetry

$$F(g) - g/2 = F(-g) + g/2. \tag{6}$$

This symmetry is also present in MF-RP model associated with RRG (see Sec. 7). It corresponds to the trajectory shown by a red vertical dashed line on the phase diagram of Fig. 1. This trajectory passes through the tri-critical point $\gamma = 4, p = 1$ on the $(\gamma, p)$ plane where the localized, multifractal and weakly-ergodic phases merge together [47]. The dynamical analysis of this paper shows that in this very point the merging of the stretch-exponential and the frozen-dynamics phases also occurs.

## 1.3 Finite-size multifractality

Our analytical approach based on the Wigner-Weisskopf approximation [30] allows us to find the *finite-size deformations* of the phase diagram. In the inset of Fig. 1 we show that a "finite-size multifractal phase" (FSMF) appears in the vicinity of the tricritical point just filling in the emerging gap between the localized and weakly-ergodic phases. The width of the gap in terms of the diagonal disorder strength $W$ is given by:

$$\frac{\Delta W}{W_c} \equiv \frac{(W_{AT} - W_{ET})}{W_c} \sim \ln K \frac{\ln \ln N + c}{2 \ln N}, \tag{7}$$

where $c \sim 1$, $W_c$ is the localization transition point in the thermodynamic limit and $W_{AT}$ and $W_{ET}$ are the finite-size apparent transition points for the localization and the ergodic transitions, respectively. This allows to define the characteristic length:

$$L_{\mathrm{MF}} = \ln N_{\mathrm{MF}} \sim \frac{\ln(|1 - W/W_c|^{-1})}{|1 - W/W_c|}, \tag{8}$$

---

[1]Here we call wave-function "weakly ergodic" if it occupies a finite fraction of the total Hilbert space. Such states play an important role in several recent works [16, 23, 24, 49–53].

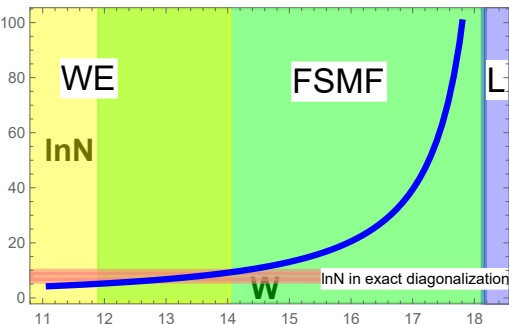

Figure 2: (Color online) **The logarithm of multifractal length** $L_{\mathrm{MF}}$ **for** $K = 2$ **RRG** given by the solution to Eq. (97) as a function of disorder strength $W$. The system sizes accessible to conventional exact diagonalization set the border line $W \sim 12-14$ between the approximately multifractal eigenfunction statistics (FSMF) and the weakly-ergodic (WE) one in the thermodynamic limit.

such that for $1 \ll N \ll N_{\mathrm{MF}}$ (at fixed disorder strength $W < W_c$) one observes approximately a multifractal eigenfunction statistics, while for $N \gg N_{\mathrm{MF}}$ it becomes weakly-ergodic.

This finding finally resolves the long-lasting dispute about the existence of the non-ergodic extended phase on RRG and gives an expression for the crossover system size $N_{\mathrm{MF}}$ that is much larger than the correlation volume at the localization transition $\ln N_c \propto |1 - W/W_c|^{-1/2}$ [21, 47, 54–56]. This new length arises because of the proximity of the tri-critical point (shown by a red point in Fig. 1) to the true multifractal phase emerging when the symmetry, Eq. (6), is infinitesimally broken. The corresponding range of disorder strength (7) (with the parameters appropriate for $K = 2$ RRG) for the currently available system sizes $N = 10^3 \ldots 10^5$ shown in Fig. 2 is in good agreement with the finite-size multifractality observed in the exact diagonalization earlier [11–13, 57].

Finally, in order to confirm the applicability of our mapping and analytically derived results for the stretch exponent $\kappa$, we studied in great detail the particular cases of LN-RP and MF-RP associated with RRG. In the first case we compute analytically the phase transition lines and the dependence of the stretch exponent $\kappa$ on the parameters $\gamma$ and $p$. In the second case, we find the function $F(g)$ from the exact theory of localization on a Cayley tree of Ref. [58] and on its basis we compute the dependence the stretch-exponent $\kappa$ on the disorder strength $W$. In both cases we found a good agreement with the exact diagonalization numerics whenever convergence to the thermodynamic limit may be reached at available system sizes or the extrapolation of data to $N \to \infty$ limit can be performed under controllable conditions.

The paper is organized as follows. In Sec. 2 we remind the criteria for the localized, multifractal and ergodic static phases in the models with the full random matrix Hamiltonians based on the principles discussed in Refs. [47, 49] and apply these principles to the MF-RP model with a generic function $F(g)$. Next, in Sec. 3 we present the main steps of the derivation of the Wigner-Weisskopf approximation for the return probability $R(t)$ and apply it in Sec. 4 to calculate the mean $\langle R(t) \rangle$ in the case of MF-RP. In the latter section we will show the general nature of the stretch-exponential decay of return probability as an emerging phenomenon and derive in Sec. 5 the transition lines between the frozen-dynamics, the stretch-exponential, and the exponential dynamical phases. Section 6 is devoted to consideration of the overlap correlation function and its relation to the stable distribution emerging beyond a certain high-frequency cutoff.

In the rest of the paper we apply the developed above general theory of MF-RP to a particular example of MF-RP that stems from the Anderson localization model on RRG (Secs. 7.1, 7.2) and inherits the symmetry Eq. (6) from the corresponding Cayley tree. More precisely, we re-

late the properties of the function $F(g)$ to the eigenvalue $\varepsilon_\beta$ of the linearized transfer-matrix operator for the Cayley tree and to the corresponding Lyapunov exponents (Secs. 7.3, 7.4). In the end of the paper we present the results of extensive numerical simulations for RRG and the corresponding LN-RP model with a symmetry parameter $p = 1$ (Sec. 8).

## 2 Localization, ergodic and fully-weakly ergodic transitions

Before studying the dynamical phases of the model (3), (4) we briefly review its static phases shown in Fig. 1.

In Refs. [47,49] we formulated the criteria for the phase transitions seen in the eigenfunction statistics of full random matrices. We identified localized, multifractal, weakly-ergodic and fully-ergodic phases and the transitions between them: Anderson localization transition (AT), ergodic transition (ET) between the multifractal and ergodic phases and the fully-weakly ergodic (FWE) transition between the the fully- and the weakly-ergodic phases. Remarkably, all the corresponding criteria can be written in a universal way using the properly defined averages of the off-diagonal matrix elements $H_{nm}$ and comparing them with the mean level spacing $\delta = E_{BW}/N$.

- Localization transition:

$$\langle |H_{nm}| \rangle_{E_{BW}} = C \int_0^\infty N^{F(g)-g/2} dg = \delta \,, \tag{9}$$

- Ergodic transition

$$\langle |H_{nm}|^2 \rangle_{E_{BW}} = C \int_0^\infty N^{F(g)-g} dg = \delta \,, \tag{10}$$

- FWE transition

$$|H_{nm}|^2_{\text{typ}} = \exp\langle \ln |H_{nm}|^2 \rangle = \delta \,, \tag{11}$$

where $\langle ... \rangle_{E_{BW}}$ denotes the upper cutoff at $|H_{nm}| \sim E_{BW}$ , and $C$ is the normalization constant

$$C^{-1} = \int_{-\infty}^\infty N^{F(g)} dg \,. \tag{12}$$

The quantities $|H_{nm}|$, its upper cutoff and $\delta$ are measured in units of the total spectral bandwidth $E_{BW}$.

In the saddle-point approximation the integrals (9) and (10) are dominated by $g = \max(g_{1/2}, 0)$ and $g = \max(g_1, 0)$, respectively, while the normalization integral (12) is given by $g = g_0$. Here we denote as $g_\beta$ the solution of equation

$$F'(g) = \beta \ . \tag{13}$$

The corresponding phase diagram for the LN-RP in $(\gamma, p)$ plane (first obtained for this model in Ref. [47]) is shown in Fig. 1 where the blue, green, and red solid lines show the corresponding transition points for the Anderson

$$\gamma_{AT} = \begin{cases} \frac{4}{2-p}, & p < 1 \\ 4p, & p \geq 1 \end{cases} \,, \tag{14}$$

ergodic

$$\gamma_{ET} = \begin{cases} \frac{1}{1-p}, & p < 1/2 \\ 4p, & p \geq 1/2 \end{cases} \,, \tag{15}$$

and fully-weakly ergodic transitions

$$\gamma_{FWE} = \frac{1}{1+p} \ . \tag{16}$$

## 3 Wigner-Weisskopf approximation.

In order to complement the above static phase diagram by the dynamical phases, we turn in this section to the main goal of this paper: the calculation of return probability in a generic MF-RP model exhibiting an emergent stretch-exponential decay.

We start by the description of the Wigner-Weisskopf (WW) approximation for the quantum dynamics [59] following the rout of Ref. [30].

In this approach the time-dependent Schroedinger equation:

$$i\,\partial_t \psi^{(n)}(i,t) = \sum_{ij} H_{ij}\,\psi^{(n)}(j,t), \quad \psi^{(n)}(i,t=0) = \delta_{ni}\,, \tag{17}$$

with the initial population only on site $n$ is solved approximately as follows. As the first step we introduce the "hopping representation":

$$\Phi^{(n)}(i,t) = \psi^{(n)}(i,t)\,e^{i\varepsilon_i t}\,, \tag{18}$$

where $\varepsilon_i = H_{ii}$ and the r.h.s. contains only off-diagonal matrix elements:

$$i\partial_t \Phi^{(n)}(i,t) = \sum_{j\neq i} H_{ij}\,\Phi^{(n)}(j,t)\,e^{i(\varepsilon_i-\varepsilon_j)t}\,. \tag{19}$$

Now $\Phi^{(n)}(i \neq n, t)$ is expressed through $\Phi^{(n)}(n,t)$ in the first order of perturbation theory:

$$\Phi^{(n)}(i \neq n, t) \approx -iH_{in}\int_0^t \Phi^{(n)}(n,\tau)\,e^{i(\varepsilon_i-\varepsilon_n)\tau}\,d\tau\,, \tag{20}$$

while the equation for $\Phi^{(n)}(n,t)$:

$$\partial_t \Phi^{(n)}(n,t) = -\sum_{j\neq n} |H_{nj}|^2 \int_0^t \Phi^{(n)}(n,\tau)\,e^{i(\varepsilon_n-\varepsilon_j)(t-\tau)}\,d\tau \tag{21}$$

is solved in the Markovian approximation $\Phi^{(n)}(n,\tau) \to \Phi^{(n)}(n,t)$ justified by the strongly-oscillating integrand:

$$\ln|\Phi^{(n)}(n,t)|^2 = -\frac{1}{2}\sum_{j\neq n}|H_{nj}|^2\,\frac{\sin^2\left(\frac{\varepsilon_n-\varepsilon_j}{2}t\right)}{\left(\frac{\varepsilon_n-\varepsilon_j}{2}\right)^2}\,. \tag{22}$$

Thus the return probability is given by:

$$R(t) \equiv |\psi^{(n)}(n,t)|^2 = |\Phi^{(n)}(n,t)|^2 = e^{-t\Gamma(t)}\,, \tag{23}$$

where we approximate:

$$\Gamma(t) \approx \frac{t}{2}\sum_{j=n-n_t/2}^{n+n_t/2} |H_{jn}|^2, \quad n_t \approx \frac{4\pi}{t\,\delta(N)}\,, \tag{24}$$

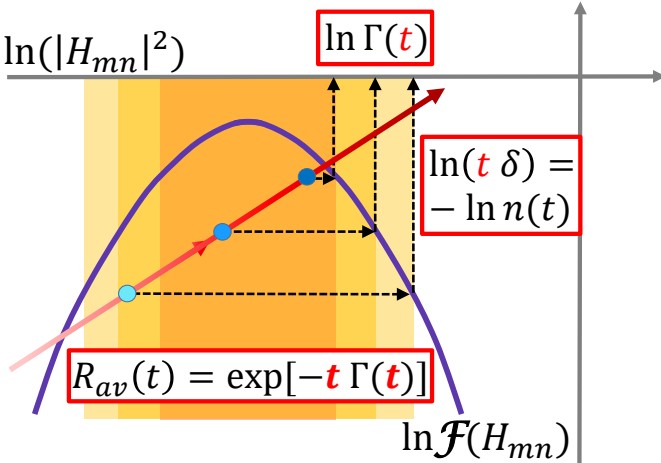

Figure 3: (Color online) **Sketch of the mechanism of slow dynamics in the MF-RP ensemble**. As time increases (the arrow of time is shown by the red diagonal line of varying intensity), the number of terms $n_t$ in the sum, Eq. (24), is shrinking. This leads to reduction of the interval of values of $|H_{nm}|$ at the tails of the distribution (shown by progressively shrinking yellow shaded areas) that one can "catch" having $n_t$ "trial events". As the decay rate $\Gamma(t)$ is determined by rare large $|H_{nm}|$, decreasing the maximal value of the set of $n_t$ random matrix elements $|H_{nm}|$ (shown by black dashed lines) results in decreasing $\Gamma(t)$ with time, i.e. the behavior of $R_{\mathrm{av}}(t)$ slower than exponential. For a distribution with no tails (e.g. when the purple parabola is very narrow) the typical maximal value of $|H_{nm}|$ is time-independent, and the dynamics of $R_{\mathrm{av}}(t)$ is exponential.

with $\delta(N)$ being the mean level spacing.

The crucial point of approximation Eqs. (23),(24) is that the dependence on time of a random decay rate $\Gamma(t)$ depends non-trivially on the interval of summation $n_t \sim 1/t$ of the random square of the matrix element $|H_{nj}|^2$, see Fig. 3. If all matrix elements $|H_{jn}|$ are typical, the sum in Eq. (24) is proportional to $|H_{typ}|^2 n_t$, that is $\Gamma(t) = const$. Thus we obtain the *exponential* decrease of return/survival probability with time. However, if the distribution of $|H_{jn}|$ is broad, then large matrix elements occur less frequently as time goes on and $n_t$ decreases. This makes the sum decaying faster than $1/t$ and $\Gamma(t)$ decreases with time (see Fig. 3).

In the next section we present the detailed derivation of this fact and show that no matter what is the function $F(g)$, the mere multifractal character of the distribution Eq. (3) makes the *average* return probability stretch-exponential in a broad interval of time. At the same time, the *typical* value of $R(t) = e^{-t\Gamma(t)}$ remains exponential, since it is determined by the typical values of $|H_{jn}|$.

Concluding this section we would like to discuss the applicability of the Wigner-Weisskopf approximation. The main approximation here is the first order of perturbation theory in Eq. (20). This approximation works well for the *full* random matrix Hamiltonian (coordination number $N \to \infty$) and fails for the sparse hopping matrix like in the initial Hamiltonian of the Anderson model on RRG (finite coordination number $K + 1$). The situation here is similar to the perturbative derivation of multifractality in the Gaussian RP model [25] which appeared to be exact [26, 27]. That is why one should first derive the RP model equivalent to the short-range sparse problem and then apply the Wigner-Weisskopf approximation to this RP model. For the latter model WW approximation works as long as $n_t \gg 1$, e.g. as long as $t \ll \delta^{-1} \sim N/E_{BW}$.

# 4 Averaging of return probability.

Given that all matrix elements $|H_{nj}|^2$ in Eq. (24) are independent random variables, the standard way of computing the distribution function $\mathcal{G}(\Gamma)$ of $-(1/t)\ln R(t) = \Gamma(t)$ is to compute first its characteristic function $\Theta(\nu) = \int e^{i\nu\Gamma} \mathcal{G}(\Gamma) d\Gamma$:

$$\Theta(\nu) = \left[ \langle e^{i\nu\frac{t}{2}|H|^2} \rangle \right]^{n_t} . \tag{25}$$

Then the average return/survival probability is

$$R_{av}(t) = \langle \exp[-t\Gamma(t)] \rangle = \Theta(\nu = it) = \langle e^{-\frac{t^2}{2}|H|^2} \rangle^{n_t} , \tag{26}$$

while the typical one is

$$R_{typ}(t) = \exp[-t\langle\Gamma(t)\rangle] = e^{-\frac{t^2}{2}n_t\langle|H|^2\rangle} . \tag{27}$$

The typical return/survival probability $R_{typ}$ is pure exponential, as $n_t \propto t^{-1}$. The situation is less trivial with the popular numerical measure of *anomalous transport*, namely the averaged survival/return probability $R_{av}(t)$. Let us consider this measure in detail. First of all we represent the average in Eq. (26) in terms of the distribution function Eq. (3):

$$\langle e^{-\frac{t^2}{2}|H|^2} \rangle = C \int dg\, e^{\ln N F(g) - \frac{1}{2}N^{2\tau - g}}, \quad t = N^\tau . \tag{28}$$

The integrand in this equation is vanishingly small at $g < 2\tau$, while at $g > 2\tau$ one can write:

$$e^{-\frac{1}{2}N^{2\tau - g}} \approx 1 - \frac{1}{2}N^{2\tau - g}.$$

Thus we obtain:

$$\langle e^{-\frac{t^2}{2}|H|^2} \rangle = C \int_{2\tau}^\infty N^{F(g)}\, dg - \frac{C}{2}\int_{2\tau}^\infty N^{F(g)+(2\tau - g)}\, dg . \tag{29}$$

Here we consider the case when:

$$2\tau < g_0 . \tag{30}$$

As we will see below this time interval is relevant for all dynamical phases.

Eq. (30) implies that (i) the first integral in Eq. (29) is dominated by the saddle-point $g = g_0$ and gives the unity with polynomial corrections of the order of $N^{F(2\tau)} \ll 1$, while (ii) in the second integral the main contribution comes either from the vicinity of the lower cutoff $g = 2\tau$, when the saddle point $g = g_1 < 2\tau$ is outside of the region of integration, or from the point $g = g_1$ itself.

The latter case, $g_1 > 2\tau$, recovers the standard diffusion, $R_{av}(t) = e^{-\Gamma_{av}t}$, with the Fermi Golden rule decay rate

$$\Gamma_{av} = \frac{n_t}{t} \ln\langle e^{-\frac{t^2}{2}|H|^2} \rangle \sim \frac{N^{F(g_1)-g_1}}{\delta(N)} = \frac{\langle|H|^2\rangle}{\delta(N)} . \tag{31}$$

The polynomial correction $N^{F(2\tau)-2\tau}$ to $\Gamma_{av}$ is subleading in this case due to the property (13) of $g_1$:

$$\max_g [F(g) - g] = F(g_1) - g_1 .$$

In the more interesting case when the saddle point $g = g_1 < 2\tau$ is outside of the region of integration in (29), the return probability is governed by the boundary term

$$-\ln R_{av}(t) \sim [\delta(N)]^{-1} N^{F(2\tau)-\tau}, \quad (g_0 > 2\tau > g_1). \tag{32}$$

At this point it is clear that the dynamics of $R_{av}(t)$ separates in three intervals, see Fig. 4

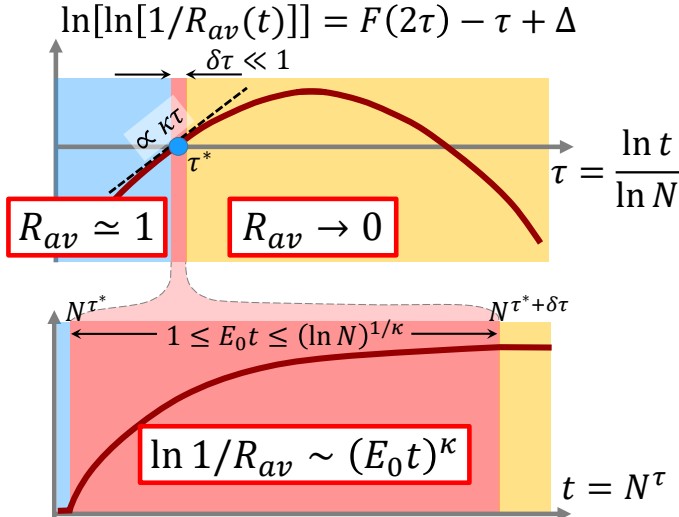

Figure 4: (Color online) **Emergent stretch-exponential dynamics in the MF-RP ensemble**. The dynamics of return probability given by Eq. (32) is divided into three regions: **(blue)** where the exponent in Eq. (32) is negative and $R_{av}(t) \simeq 1 - N^{-c}$, $c > 0$; **(yellow)** where the exponent is positive but Eq. (32) loses its validity due to the saturation of $R_{av}(t) \simeq N^{-D_2}$ which is beyond WW approach, and **(pink)** the close vicinity $\delta\tau \sim \ln\ln N / \ln N$ of the point $\tau = \tau^*$ where the exponent in (32) crosses zero. This region, being a small interval $\delta\tau \ll 1$ in $\tau$ (upper panel) blows up into an extensive interval $1 \lesssim (E_0 t)^\kappa \lesssim \ln N$, Eq. (41) in real times $t = N^\tau$ (lower panel).

**(i)** At small times (shown by light blue) $F(2\tau) - \tau + \Delta < 0$, where the parameter $\Delta$ is defined as

$$\delta(N) = E_{BW}/N \sim N^{-\Delta}, \tag{33}$$

the evolution is perturbative as $R_{av}(t) = 1 - N^{-[\tau - F(2\tau) - \Delta]} \simeq 1$

**(ii)** At large times $F(2\tau) - \tau + \Delta > 0$ (yellow), Eq. (32) shows very fast evolution, but it will be immediately terminated at $F(2\tau) - \tau + \Delta \sim \ln\ln N / \ln N$ by the finite-size saturation of return probability of the order of $-\ln R_{av}(t \to \infty) \sim \ln N$ (see below), while

**(iii)** The range of non-perturbative dynamics of our interest (shown by pink) is located in the vicinity $\delta\tau \ll 1$ of the point $\tau = \tau^*$ given by the solution of the equation

$$F(2\tau^*) - \tau^* + \Delta = 0. \tag{34}$$

Here the crucial step in understanding of the *emergent stretch-exponential behavior* is the expansion in $\delta\tau = \tau - \tau^* \ll 1$ of $F(2\tau)$ in Eq. (32) in the vicinity of $\tau = \tau^*$ (upper panel of Fig. 4)

$$
\begin{aligned}
n_t \ln\langle e^{-\frac{t^2}{2}|H|^2}\rangle &\sim N^\Delta N^{F(2\tau^*) - \tau^*} N^{[2F'(2\tau^*) - 1]\delta\tau} \\
&= (N^{-\tau^*} t)^\kappa,
\end{aligned} \tag{35}
$$

where

$$\kappa = 2F'(2\tau^*) - 1. \tag{36}$$

The expression (35) defines the only characteristic energy scale $E_0$ separating the three dynamical regions discussed above:

$$E_0 \sim N^{-\tau^*}, \tag{37}$$

where $\tau^*$ is the solution to Eq. (34). In ergodic phases $E_0$ is of the order of the total band-width $E_0 \sim E_{BW} = N\,\delta(N)$, while in the multifractal phase $E_0$ has a physical meaning of a width of the *mini-band* in the local spectrum. In this case it decreases polynomially with increasing $N$, i.e. $\tau^* > 0$.

In order to find the scaling exponent $\tau^*$ one should use the knowledge about the scaling of the total bandwidth $E_{BW} \sim N^{1-\Delta}$. It is well-known [25,47,53] that in the non-ergodic (e.g. localized and multifractal) phase $E_{BW} = W/2 \sim N^0$, so that $\Delta = 1$. However, in the weakly- or fully-ergodic phases $\Delta < 1$. Indeed, using the fact that in this case $E_0 \sim E_{BW}$ one obtains for ergodic phases $\Delta = 1 + \tau^*$, where $-1 < \tau^* < 0$. Therefore:

$$\Delta = \begin{cases} 1 + \tau^*, & \text{ergodic phase, } \tau^* < 0 \\ 1, & \text{non-ergodic phases} \end{cases}. \tag{38}$$

Now, Eq. (34) can be written in a general form:

$$F(2\tau^*) = \begin{cases} -1, & \text{ergodic phase} \\ \tau^* - 1, & \text{non-ergodic phases} \end{cases}. \tag{39}$$

For us relevant is the *smaller* root of Eq. (39) with $2\tau^* < g_0$ where the dynamics is not yet saturated.

Eqs. (36), (37), (39) uniquely determine the stretch exponent $0 < \kappa < 1$ in the survival/return probability:

$$R_{\text{av}}(t) \approx \exp[-(E_0 t)^\kappa]. \tag{40}$$

This is the main result of the paper.

To find the true region of its validity we note that in a finite system there is a saturation of $R_{\text{av}}(t) = \sum_\alpha |\psi_\alpha(n)|^4 \sim N^{-D_2}$ at large times expressed in terms of the complete set of eigenstates $\psi_\alpha(n)$ of the MF-RP problem. This saturation is beyond the Wigner-Weisskopf approximation and this reduces its validity range from $t \lesssim 1/\delta(N)$ to

$$\ln(1/R_{\text{av}}) < D_2 \ln(N), \quad \Leftrightarrow \quad 1 \lesssim t E_0 \lesssim [\ln N]^{\frac{1}{\kappa}}. \tag{41}$$

It is important to note that this domain of validity corresponds to $0 < \delta\tau \lesssim \kappa^{-1} \ln \ln N / \ln N \ll 1$ (upper panel of Fig. 4). Thus despite the width of the interval of $E_0 t$ diverges as $N \to \infty$ (lower panel of Fig. 4), the width of $\delta\tau$ shrinks to zero, hence the expansion of $F(g)$ around $g = 2\tau$ becomes exact in the thermodynamic limit. This is the consequence of the "multifractal" character of the distribution Eq. (3). It is generic to any distribution of this type, no matter what the function $F(g) \sim N^0$ of $g \sim N^0$. Thus we have shown that the stretch-exponential behavior of the survival/return probability is an *emergent property* of a large "multifractal" RP random matrices when the thermodynamic limit $N \to \infty$ is taken before $E_0 t \to \infty$.

# 5 Exponential/stretch-exponential and frozen-dynamics transitions

In order to obtain the stretch-exponential behavior Eqs. (40) it is crucially important that the lower cut-off $2\tau \approx 2\tau_*$ in the integrals in Eq. (29) is larger than the saddle point $g_1$ in the second integral (but smaller than the saddle point $g_0$ in the first one). If this is not the case then the survival probability $R_{\text{av}}(t)$ is *pure exponential*, $\kappa = 1$ with the decay rate (31) given by the Fermi Golden rule.

Thus the transition between the exponential and the stretch-exponential dynamics happens at:

$$2\tau^* = g_1, \quad \Rightarrow \quad F(g_1) - g_1/2 = -\Delta. \tag{42}$$

Eq. (42) is general for any MF-RP model with the function $F(g)$ obeying only the normalization condition $F(g_0) = 0$. It can be rewritten in a physically transparent form similar to (9), (10), (11). For this we note that the LN-RP distribution with the parabolic function $F(g)$ obeys a special symmetry:

$$F(g_1) - g_1/2 = F(g_0) - g_0/2 \,. \tag{43}$$

This symmetry includes the symmetry, Eq. (6), as a particular case, but it is wider than that which allows only one value of $p = \langle \delta g^2 \rangle / (2\langle g \rangle) = 1$. However, it does not cover all the possible functions $F(g)$.

Using the normalization $F(g_0) = 0$, the symmetry (43), and the definition $H_{typ} = N^{-g_0/2}$, we reduce the condition, Eq. (42), to the one covering *any* LN-RP and RRG

$$H_{typ} = \delta(N) \,. \tag{44}$$

We believe that this condition is the basic one for all the physically-motivated applications. The physics behind it is that at $H_{typ} \gg \delta(N)$ all the states are typically hybridized and the Fermi Golden Rule holds leading to the pure exponential dynamics. In contrast, at $H_{typ} \ll \delta(N)$ only atypically large couplings with $|H_{mn}| > \delta(N)$ can hybridize the corresponding states, all other couplings may be safely neglected. They constitute an effective sparse matrix of couplings, and so the stretch-exponential behavior of $R_{av}(t)$ emerges exactly at the point where the effective matrix of couplings becomes sparse (see [60]).

Now let us consider the condition that the stretch-exponent $\kappa$ turns to zero. At this point the dynamics of survival probability is at most some power law of time. Such an ultra-slow dynamics is essentially a *glassy* behavior which we will refer to as the *frozen-dynamics* phase.

Its physical meaning is related to the *mobility edge*. Indeed, according to Eq. (52) the return probability saturates at large $t$ when *different* states close in the energy do not overlap in an observation point of an infinite system. Obviously, this happens when the states are localized. However, it may happen also if all *extended* states which are close in the energy do not populate the observation point. The reason is that there is a *remote in energy state* beyond the mobility edge that is localized exactly at this point. Then by orthogonality of states all other (extended) states must have a population hole at this site [61]. This is a somewhat exotic situation, as the localized state must be localized on one single site in the limit $N \to \infty$, otherwise the orthogonality condition may be met by matching of phases of wave function (or its signs). However, this is known to happen in the RP models [32].

According to Eq. (36) the stretch-exponent $\kappa$ turns to zero when:

$$\left( \frac{dF}{dg} \right)_{g=2\tau^*} = \frac{1}{2}, \;\; \Rightarrow \;\; 2\tau^* = g_{1/2} \,. \tag{45}$$

Using Eq. (34) one can express the condition for the transition to the frozen-dynamics phase in the form very similar to the Anderson transition (9)

$$\langle |H| \rangle = \delta(N) \,, \tag{46}$$

with the only difference of the absence of the upper cutoff at $|H_{mn}| \sim O(N^0)$ which is present in (9). This immediately means that F/SE-transition coincides with the Anderson transition as soon as $2\tau^* = g_{1/2} \geq 0$, i.e. when the cutoff is not important. On the other hand, $\tau^* > 0$ implies the multifractal phase. We conclude, therefore, that *in a generic MF-RP model* the F/SE transition is not possible *inside* the multifractal phase. Indeed, it coincides with the Anderson localization transition when the multifractal phase is present. However, there is no prohibition for the F/SE transition to be inside the weakly-ergodic phase, and, indeed, it happens there (see Fig. 1).

The fact that the criteria of *all* the phase transitions can be presented in the common form (9), (10), (11), (44), and (46) allows to establish a general sequence of transitions as disorder increases. By writing the corresponding general inequalities for averages of $|H_{nm}|$ and $|H_{nm}|^2$, we conclude that the SE/E transition (44) has to appear at smaller disorder compared to the F/SE transition (46), while both of them should be located between the Anderson (9) and the fully-weakly ergodic transition (11), irrespectively of the location of ergodic transition.

As an example of application of the criteria (9), (10), (11), (44), and (46) we compute the transition lines in the LN-RP model suggested in [47] and further developed in [35, 60].

For this model the function $F(g)$ is parabolic:

$$F(g) = -\frac{(g-\gamma)^2}{4p\gamma}, \tag{47}$$

and one can easily find

$$g_1 = (1-2p)\gamma, \quad g_{1/2} = (1-p)\gamma. \tag{48}$$

The smaller of the two solutions of Eq. (39) corresponding to this function is:

$$2\tau^* = \begin{cases} \gamma - 2\sqrt{\gamma p}, & \text{ergodic phase} \\ \gamma(1-p) - \sqrt{4p\gamma - \gamma^2 p(2-p)} & \text{multifractal phase} \end{cases}. \tag{49}$$

Then solving the equation $2\tau^* = g_1$ one obtains for the E/SE transition two different expressions depending on whether the transition happens inside the weakly-ergodic or inside the multifractal phases:

$$\gamma_{E/SE} = \begin{cases} \frac{1}{p}, & \text{ergodic phase} \\ 2, & \text{multifractal phase} \end{cases}. \tag{50}$$

Analogously, from $2\tau^* = g_{1/2}$ one obtains the line of transition to the frozen-dynamics phase:

$$\gamma_{FD} = \begin{cases} \frac{4}{p}, & \text{ergodic phase} \\ \frac{4}{2-p}, & \text{multifractal phase} \end{cases}. \tag{51}$$

Eqs. (50),(51) should be complemented by the lines of the Anderson localization (14), the ergodic transition (15), and the transition between the weakly-ergodic and the fully ergodic phases (16) first obtained for this model in Ref. [47]. All these transition lines are shown in Fig. 1. Please notice a complexity of this diagram which possesses *four multi-critical points* and *seven* different phases, including three types of weakly-ergodic phases and two types of multifractal phases.

Notice also that the phase with frozen dynamics appears *inside* the weakly-ergodic phase, and the phase with the stretch-exponential dynamics appears both inside the weakly-ergodic and inside the multifractal phase. This implies that the classification by *dynamical phases* reflects different, complementary features of eigenfunction statistics compared to the classification by the extent of *ergodicity violation*. While the localized, multifractal, weakly- and fully-ergodic phases are classified according to the fractal dimension of the support set of a *single* eigenfunction and the fraction of the occupied sites in it, the classification by dynamical phases accounts for the *correlation* between eigenfunctions of different energies.

## 6 Overlap correlation function and Levy stable distribution

The survival/return probability $R_{av}(t)$ is the Fourier-transform of a correlation function (discussed, e.g., in Ref. [62])

$$C(\omega) = \left\langle N^{-1} \sum_{\alpha,\beta,r} |\psi_\alpha(r)|^2 |\psi_\beta(r)|^2 \delta(\omega - E_\alpha + E_\beta) \right\rangle, \tag{52}$$

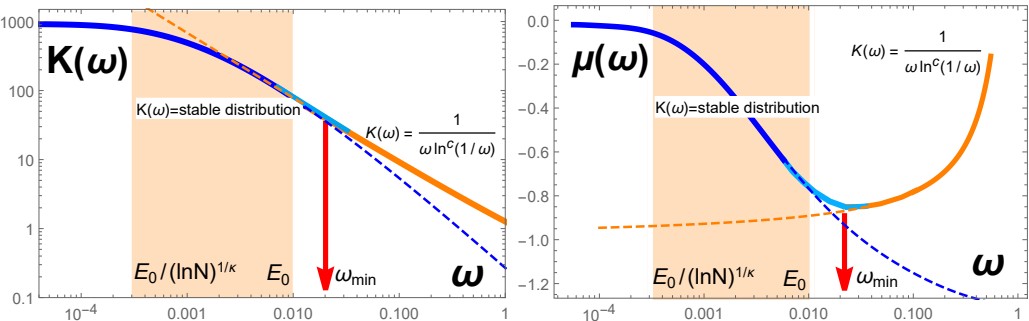

Figure 5: (Color online) **The emergent Levy stable distribution.** The region of $\omega$ (shown by the orange shadow) on the left from the minimum in $\mu(\omega)$ gives the main contribution to the Fourier-transform of $K(\omega)$ to obtain the stretch-exponential behavior of $R_{av}(t)$ at $E_0^{-1} \lesssim t \lesssim E_0^{-1} (\ln N)^\kappa$. The corresponding parts of the curves for $K(\omega)$ and $\mu(\omega)$ (blue solid lines) are the part of the symmetric about $\omega = 0$ Levy $\alpha$-stable distribution with $\alpha = \kappa$ and its log-log derivative, respectively (their continuations are shown by the blue dashed lines) . The parts of the curve $K(\omega)$ and $\mu(\omega)$ on the right of the minimum $\omega_{min} \sim E_0$ of $\mu(\omega)$ (orange solid lines) are $K(\omega) \sim \omega^{-1} (\ln(1/\omega))^{-c}$, which was analytically derived for RRG in Ref. [21], and its log-log derivative (their continuations are shown by the orange dashed lines). This part of $K(\omega)$ is relevant for the Fourier-transforming at shorter times $E_{BW}^{-1} \lesssim t \lesssim E_0^{-1}$.

where $\psi_\alpha(r)$ is the eigenfunction corresponding to eigenenergy $E_\alpha$ at a site $r$. To show this one expands the wave function $\psi^{(n)}(r,t) = \sum_\alpha a_\alpha \psi_\alpha(r) e^{-iE_\alpha t}$ in the eigenbasis. If the coefficients are chosen as $a_\alpha = \psi_\alpha(n)$, the completeness imposes the correct initial conditions $\psi^{(n)}(r, t=0) = \delta_{r,n}$. Then one immediately obtains

$$R_{av}(t) \equiv |\psi^{(n)}(n,t)|^2 = \int d\omega \, C(\omega) e^{i\omega t} \; . \tag{53}$$

In order to eliminate the effect of level repulsion, we normalize $C(\omega)$ by the two-level correlation function $Q(\omega) = \left\langle N^{-1} \sum_{\alpha,\beta} \delta(\omega - E_\alpha + E_\beta) \right\rangle$ and define the *eigenfunction overlap* correlation function:

$$K(\omega) = \frac{C(\omega)}{Q(\omega)} \; . \tag{54}$$

The functions $C(\omega)$ and $Q(\omega)$ both have a correlation hole at $\omega \ll \delta$ due to level repulsion, while for $\omega \gg \delta$ the function $Q(\omega) = \delta^{-1}$ is a constant. Therefore, for exact diagonalization at a finite $N$ the function $K(\omega)$ is preferred over $C(\omega)$, as it allows to separate the effects of eigenfunction statistics from the trivial level repulsion. This poses an advantage, as it extends the region of the stretch-exponential behavior to larger times compared to direct computation of survival probability in the time domain.

Taking the inverse Fourier transform of Eq. (53) one can find $K(\omega)$ at $\omega \gg \delta$:

$$K_{av}(\omega \gg \delta) \sim \delta \int dt \, R_{av}(t) e^{-i\omega t} \; . \tag{55}$$

In order to get a more detailed information we also consider the log-log derivative of $K(\omega)$:

$$\mu(\omega) = \omega \, \partial_\omega \ln K(\omega) . \tag{56}$$

It gives the "running power" of the locally power-law function $K(\omega)$.

Taking into account a well-known characteristic function of the Levy $\alpha$-stable distribution:

$$\chi_\alpha(t) = \exp\left[i\mu t - |bt|^\alpha \left(1 - i\beta \operatorname{sign}(t) \tan\left(\frac{\pi\alpha}{2}\right)\right)\right], \tag{57}$$

we conclude from Eqs. (55),(40) that the stretch-exponential $R_{\mathrm{av}}(t)$ corresponds to the symmetric Levy $\alpha$-stable distribution:

$$K(\omega) = f(\omega; \alpha, \beta, \mu, b), \tag{58}$$

with $\alpha = \kappa$, the skewness parameter $\beta = 0$, the shift parameter $\mu = 0$ (the distribution is symmetric about $\omega = 0$) and the scale parameter $b = E_0$. This result is valid for $\omega$ corresponding to the region of validity of SE, Eq. (41):

$$E_0/(\ln N)^{\frac{1}{\kappa}} \lesssim \omega \lesssim E_0. \tag{59}$$

For $E_{BW} \gtrsim \omega \gtrsim E_0$ the main contribution to the Fourier-transform of $R_{\mathrm{av}}(t)$ comes from $t \sim \omega^{-1}$:

$$K(\omega) \sim \frac{R_{\mathrm{av}}(t \sim |\omega|^{-1})}{|\omega|}. \tag{60}$$

This is consistent with the analytical result of Ref. [21] for RRG:

$$K(\omega) \sim \frac{1}{|\omega| \ln^c(1/|\omega|)}, \quad c \sim 1/2. \tag{61}$$

The parts of $K(\omega)$ corresponding to the Levy $\alpha$-stable distribution and Eqs. (60),(61) are shown in Fig. 5 by the blue and the orange solid lines, respectively. A superficial look at $K(\omega)$ could give a wrong impression that the entire falling part of $K(\omega)$ is described by $K(\omega) \propto |\omega|^{-1}$, possibly with some logarithmic corrections [13,21,25]. To see that this is not so, one has to plot the log-log derivative of $K(\omega)$, Eq. (56), shown on the lower panel of Fig. 5. On this plot the part corresponding to the log-log derivative of the Levy $\alpha$-stable distribution (blue solid line) matches with the log-log derivative of Eq. (61) (orange solid line) near the minimum of the log-log derivative $\mu(\omega)$ of $K(\omega)$. Either of the two parts of $\mu(\omega)$ do not have a minimum and their continuations (shown by the dashed lines of the corresponding color) tend asymptotically to $-(1+\kappa)$ and $-1$, respectively. However, at the matching point a minimum in $\mu(\omega)$ emerges with the minimal value of $\mu(\omega)$ being $|\mu_{\min}| \leq 1$. Thus the minimum in $\mu(\omega)$ is a natural marker of termination of the emergent Levy $\alpha$-stable distribution in $K(\omega)$.

The numerical results for $\mu(\omega)$ shown in Appendix A (see Fig. 9) demonstrate that the minimum in $\mu(\omega)$ indeed exists in the weakly-ergodic, stretch-exponential phase of LN-RP model with $p = 1$, as well as on RRG. However, in the multifractal, stretch-exponential phase of LN-RP model at $p = 1/2$ there is, instead, a shoulder. This signals again of the two different regimes in $K(\omega)$, the one that is described by the Levy $\alpha$-stable distribution and another one, which, however, is different from Eq. (61).

## 7 Correspondence between Anderson model on RRG and MF-RP

In the rest of the paper we apply the general idea of mapping to MF-RP model to a particular example of the Anderson tight-binding model on a random regular graph (RRG). We argue that a RP random matrix model with a properly chosen "multifractal" distribution of i.i.d. off-diagonal (hopping) matrix elements, is *in the same universality class* as RRG. The corresponding function $F(g)$ in the distribution, Eq.(3), is determined by the eigenvalue $\varepsilon_\beta$ of the linearized transfer-matrix operator for the corresponding Cayley tree (CT) with the branching number

$K = m-1$, where $m$ is the coordination number of RRG. It obeys a special symmetry that stems from the basic Abou-Chacra-Thouless-Anderson (ACTA) symmetry $\varepsilon_\beta = \varepsilon_{1-\beta}$ [58]. It is this symmetry which ensures that the Anderson transition point in the RP ensemble equivalent to RRG is in fact the tri-critical point of a more generic RP model where the localized, multifractal and ergodic phases meet together [47]. This is the reason for existence of a transient finite-size multifractal phase on RRG [11,12] with ergodicity of states restored in the thermodynamic limit $N \to \infty$ [14,21].

However, the extended states on RRG are not *fully ergodic* like in the classic Wigner-Dyson random matrices. We show that the survival probability $R_{av}(t)$ in the RP ensemble corresponding to RRG is *stretch-exponential* in the entire extended phase [47], like it was earlier conjectured in Ref. [23]. This statement rests on the proposed equivalence between the RP and RRG and on the ACTA symmetry and is valid for any distribution of on-site energy with no fat tails and for any branching number $K$ of RRG.

We show that effective RP ensemble which is equivalent to RRG is of the form Eq. (3). However, it must also respect the ACTA symmetry (6):

$$\epsilon_\beta = \epsilon_{1-\beta}, \iff F(g) - g/2 = F(-g) + g/2. \tag{62}$$

We will show below that this symmetry enforces that the F/SE transition on RRG coincides with the Anderson localization transition and the SE/E transition coincides with the point where the Lyapunov exponent $\lambda_{typ} = -(1/2)\partial_\beta \epsilon_\beta|_{\beta=0}$ on the corresponding Cayley tree takes its minimal value $\lambda_{typ} = (1/2) \ln K$ characterizing the fully-ergodic phase [63]. For the conventional Anderson model on a CT with one orbital per site this value is reached only in the limit of vanishing disorder, so that the entire weakly-ergodic phase on RRG is characterized by a stretch exponential dynamics [2]. Furthermore, we will show that SE phase on RRG takes exactly the place of MF phase on a corresponding Cayley tree. Thus when passing from the finite CT to the RRG (loosely speaking "by connecting the leaves"), the multifractal phase is transformed [20,64] into the weakly-ergodic phase with stretch-exponential dynamics. The latter is thus the *remnant* of the finite-size multifractality discovered in Refs. [11,12].

We would like to emphasize that the form of the distribution $\mathcal{F}(x = H_{nm})$ corresponding to RRG is strongly disorder-dependent and is not the log-normal [47] in general. Thus the details of the phase diagram do depend on the precise form of the dependence of $\epsilon_\beta$ on $\beta$ and the function $F(g)$ that follows from it. However, the *topology* of the phase diagram is generic and is well represented by the "symmetric" log-normal RP model of Fig. 1 with $p = 1$.

## 7.1 Mapping of tight-binding Anderson model on RRG onto RP model with long-range hopping

Before justifying the mapping of the Anderson localization model with *nearest neighbor* hopping on RRG to the Rosenzweig-Porter random matrix model with *infinite-range* hopping we review the basic facts about RRG.

The random $K$-regular graph (RRG) of $N$ sites is a graph in which any site is connected to $K + 1$ other sites in a random way. The hopping amplitude $V = 1$ between nearest neighbor sites is fixed for all links. The Anderson localization model on RRG adds a random potential $\varepsilon_n$ fluctuating independently at any site $n$ with the site-independent distribution $\mathcal{P}(\varepsilon)$ characterized by the variance $\langle \varepsilon_n^2 \rangle \sim W^2$.

---

[2] For the sigma-model (SM) on CT which corresponds to the *infinite* number of orbitals per site, the limit $\lambda_{typ} = (1/2) \ln K$ is reached at a finite "conductance" $\mathfrak{g}$, while $\lambda_{typ} \to 0$ in the "clean" limit $\mathfrak{g} \to \infty$. It happens because of the lack of ballistic regime and the corresponding "kinetic" contribution to the spectral bandwidth in the sigma-model. This leads to the ergodic transition at a finite $\mathfrak{g}$ corresponding to $\lambda_{typ} = (1/2) \ln K$ [20]. On the SM RRG this point corresponds to the WE-FE transition which coincides with the SE-E transition.

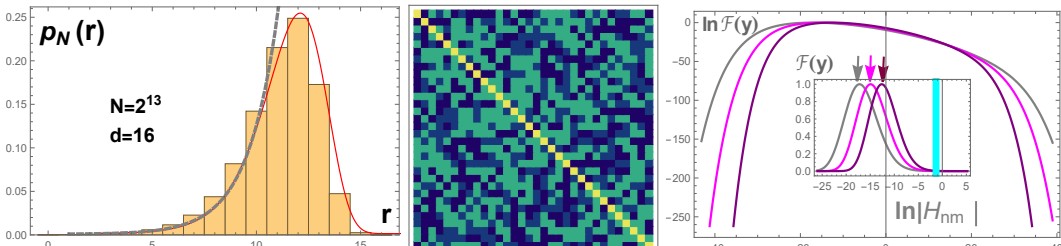

Figure 6: (Color online)(Left panel)**The distribution of distances on RRG**. The probability of large distances $r$ grows exponentially like $K^r$ (gray dashed line) until it reaches the maximal value $r_{max} \approx d \approx \ln N / \ln K$. (Middle panel)**The matrix of distances on RRG.** Large distances $r \approx d$ constitute a finite fraction of all the distances between two vertices. In each row of the distance matrix on RRG there are $\sim N$ matrix elements (shown in dark blue on right panel) which correspond to pairs of vertices with large distances $r \approx d$ between them. These "halo" sites are connected by multiple loops on RRG. The sites with small distances $r \lesssim d$ can be considered as a "medium" through which an effective hopping between the "halo" sites occurs in the high-order of perturbation theory in the nearest neighbor hopping on RRG. These sites form an almost loop-less, tree structure which allows to compute the distribution function of i.i.d. effective large-distance hopping matrix elements between the "halo" sites. We conjecture that the Rosenzweig-Porter random matrix model obtained by completing the random matrix of effective hopping over the "halo" sites to full matrix (complete graph) captures all the essential large-distance features of the Anderson model on RRG. (Right panel) **Logarithm of distribution of** $\ln |H_{nm}|$ at $N = 2048$ (purple), 8192 (magenta), 32768 (gray) for $W = 12$. The arrows show the typical value of $\ln |H_{nm}|$. The inset shows $\mathcal{F}(y = \ln |H_{nm}|)$ with the cyan strip indicating the region of $\ln |H_{nm}|$ which is relevant for the eigenfunction statistics and dynamical properties. All energies are measured in units of the total spectral bandwidth $E_{BW}$.

RRG is known to have locally a Cayley-tree structure with the branching number $K$. However, in contrast to a finite Cayley tree which is a *hierarchical graph* growing from a certain *root* onwards, any point of RRG can be considered as a root of a local Cayley tree. This is because RRG has loops (of which overwhelming majority are long loops with the length of the order of the diameter of the graph) which connect one local tree with the other thus making all points of the graph statistically equivalent.

The local tree structure and the predominance of long loops on RRG lead to the exponential growth of the distribution $p_N(r) \sim K^r / N$ of distances $r$ between pairs of vertices on this graph. This growth lasts nearly up to the maximal distance on RRG (the diameter $d \approx \ln N / \ln K$), followed by an abrupt drop to zero. The most probable distance, $r = r_0$, differs from the graph diameter $d$ only by few units and for large graphs of $N$ vertices they are approximately equal $r_0 \simeq d$ (see the left panel of Fig. 6).

Moreover, due to the exponential growth of $p_N(r)$ with $r$, in the thermodynamic limit $N \to \infty$ a finite fraction of pairs of sites on RRG is at the most probable distance, $p_{N \to \infty}(r_0) \to f > 0$. This "condensation of large distances" is the crucial point for the mapping of the Anderson model on RRG onto a RP model.

Indeed, let us take any vertex ("site") $n$ on RRG and consider the set of vertices $m(n)$ at distances $r_0 - a < |n - m(n)| < r_0$ ($a \ll r_0$) around the most probable distance $r_0$ from it. We call them the "halo" sites. The other sites $j$ at distances $|n - j| < r_0 - a$ will be referred to as the "local tree" sites. Suppose that the local tree structure of RRG with the root at the site $n$ holds strictly for all the sites $j$ (i.e. the loops are absent). Then there is one single path from the

site $n$ to any site $j$. Thus one can compute the Green's function $G_{n,j}$ as a product of the (real) single-site Green's functions $G_{l,l}$ of a Cayley tree along the path from $n$ to $j$. We assume that this property holds up to the distance $|n-j| = r_0 - a$ (with a proper choice of $a$), so that the transmission amplitude from the site $n$ to the "halo sites" $m(n)$ can be computed within this "tree approximation". In this way one finds *an effective hopping matrix element* $H_{nm} = G_{nm}$ from $n$ to $m(n)$. The matrix element $H_{nm}$ is a strongly fluctuating quantity of random sign. However, the probability distribution $\mathcal{F}(y = \ln|H_{nm}|)$ is identical for all $n, m(n)$ because of the nearly fixed distance $r_{nm} \approx r_0 \approx d$ to any of the "halo" sites. It is also independent of the initial site $n$, since all the sites on RRG are statistically equivalent. So, we obtain a random matrix ensemble with the i.i.d. distribution of the off-diagonal matrix elements $H_{nm} = G_{nm}$ fully determined by the local tree structure of RRG. However, the two "halo" sites $m_1(n_1)$ and $m_2(n_2)$ are not necessarily at the distance $\sim d$ from each other. This means that the random matrix $H_{nm}$ is *not* a full matrix representing a *complete graph*. Rather, it has a structure of a *large-distance matrix* on RRG (see middle panel of Fig. 6) with the matrix elements $H_{nm}$ between the "halo" sites shown in dark blue. The complete sub-graphs ("cliques") corresponding to this matrix have maximal sizes $\sim \ln N \ll N$. Notwithstanding this we conjecture that the scaling with $N$ of statistics of eigenfunctions and the stretch-exponential dynamics in so defined random matrix ensemble is identical to that in the *Rosenzweig-Porter* random matrix ensemble, in which *all* the off-diagonal matrix elements are i.i.d. with the distribution $\mathcal{F}(y = \ln|H_{nm}|)$. The reason for that is that even in typically full random matrix drawn from the RP ensemble these properties are determined by atypically large [3] hopping matrix elements (shown by a cyan strip in right panel of Fig. 6). Since the *relevant* large matrix elements in a RP matrix form anyway a *sparse* matrix, the sparseness of the matrix $H_{nm} = G_{nm}$ does not matter. What matters, instead, is the broad, large-deviation character of the distribution $\mathcal{F}(y = \ln|H_{nm}|)$, Eq. (3) (see right panel of Fig. 6).

We would like to note that the replacement of the short-range hopping matrix on RRG by the long-range hopping matrix in the equivalent RP ensemble is not an exact transformation. Rather, it has the meaning similar to the replacement of the sparse matrix of hopping in the three-dimensional Anderson model (3D AM) by the full matrix of classical Wigner-Dyson (WD) ensemble describing well the level- and eigenfunction statistics in 3D AM at small-energies/large times in the extended state. However, it does not apply to the "high-energy" properties like the spectral bandwidth $E_{BW}$. The latter is independent of $N$ in 3D AM and $\propto \sqrt{N}$ in the Wigner-Dyson semi-circle. The correspondence between the two models is established after the level spacing is measured in units of the mean-level spacing $\delta = E_{BW}/N$ and on the scale of energies smaller than the Thouless energy. Likewise, a proposed correspondence between the the Anderson model on RRG and the RP random matrix model requires that all the energies are measured in units of the spectral bandwidth and are much smaller than this scale.

Furthermore, in the case of long-range, semi-classical impurities in 3D metals one should distinguish between the Thouless energy ($E_{Th}$) below which WD level statistics is valid and the Ehrenfest energy ($E_{Ehr}$) above which perturbation theory works. In the interval of times $1/E_{Ehr} < t < 1/E_{Th}$ (or in the corresponding interval of energies) the dynamics is *diffusive*. In the case of RRG the role of the Ehrenfest energy is played by $E_0 \sim E_{BW}$, while the role of the Thouless energy is played by

$$E_{Th} = \frac{E_0}{(\ln N)^{1/\kappa}} \ . \tag{63}$$

In the interval of times $E_{Ehr}^{-1} \lesssim t \lesssim E_{Th}^{-1}$ the survival probability $R_{\text{av}}(t)$, Eq. (5), is *stretch-exponential* (see Eqs.(40),(41)).

---

[3] Atypically large hopping matrix elements compared to the small typical matrix elements $H_{typ} \ll \delta$ in both models (cf. Eq. (44))

As the diffusion on the Cayley tree results in the exponentially decreasing survival probability, the stretch-exponential one may be considered as a *sub-diffusion*.

## 7.2 "Multifractal" distribution of hopping

Next, we consider generic properties of the distribution function $\mathcal{F}_r(y = \ln|G_{nj}|)$ of the two-point Green's functions $G_{nj}$ on an infinite Cayley tree.

The crucial point here is that in the absence of loops the two-point cavity Green's function $G_{nj,l\to k}$ (short-hand notation $G_{nj}$) can be expressed in terms of the product of one-point cavity Green's functions $G_{p\to p'}$ (short-hand notation $G_p$) along a (unique) path of length $r = |n - j|$ that connects the points $n$ and $j$:

$$G_{nj} = \prod_{p\in\text{path } n\to j} G_p \,. \tag{64}$$

In order to proceed further we use the result of Refs. [13,22] where it is shown that the moments $I_\beta = \langle|G|^{2\beta}\rangle$ are expressed in terms of the eigenvalue $\epsilon_\beta$ of the linearized transfer-matrix operator (analytically continued from the segment $\beta = [0,1]$ to entire complex plane $\beta \in \mathbb{C}$), first introduced in the seminal work of Abou-Chacra, Thouless and Anderson (ACTA) [58] and derived in the supersymmetry framework in [65]:

$$I_\beta \propto [\epsilon_\beta]^r \,. \tag{65}$$

Then the Mellin transform yields:

$$\mathcal{F}_r(y) = \int_B \frac{d\beta}{\pi i}\, e^{-2\beta y + r\ln\epsilon_\beta} \,, \tag{66}$$

where the integration is performed over the Bromwich $\beta = c + iz$ contour which goes parallel to the imaginary axis ($\operatorname{Im}\beta \in [-\infty, +\infty]$) on the positive side of the real one ($c > 0$).

The eigenvalue $\epsilon_\beta$ obeys a celebrated ACTA symmetry [13,58]:

$$\epsilon_\beta = \epsilon_{1-\beta} \,, \tag{67}$$

and the identities:

$$\epsilon_0 = \epsilon_1 = 1, \quad \partial_\beta\epsilon_\beta|_{\beta=1} = -\partial_\beta\epsilon_\beta|_{\beta=0} \,. \tag{68}$$

As a result [13], in the limit of long path $r \gg 1$ the distribution $\mathcal{F}_r(y = \ln|G_{nj}|)$ of the product Eq. (64) has a special symmetry:

$$\mathcal{F}_r(-y)e^{-y} = \mathcal{F}_r(y)e^y \,, \tag{69}$$

which coincides with Eq. (62).

At large $r$ one can compute the Mellin transform in the saddle-point approximation:

$$\ln(\mathcal{F}_r(y)) = r\left(\ln\epsilon_\beta - \beta\partial_\beta\ln\epsilon_\beta\right)_{\beta=\beta_*} \,, \tag{70}$$

where $\beta_*$ is found from the stationarity condition:

$$\left(\partial_\beta\ln\epsilon_\beta\right)_{\beta=\beta_*(y)} = \frac{2y}{r} \,. \tag{71}$$

Eq. (71) implies that $\beta_*$ is a function of $y/r$. It follows from Eq. (70) that

$$\mathcal{F}_r(y) \sim e^{r\ln K\, F\left(-\frac{2y}{r\ln K}\right)} \,, \tag{72}$$

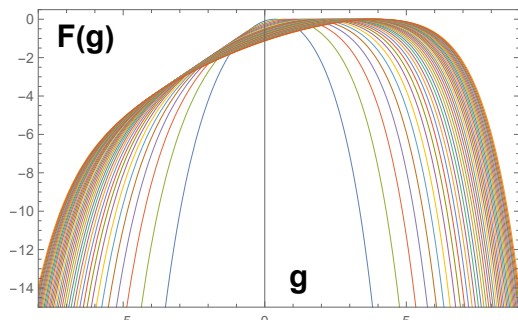

Figure 7: (Color online) **Evolution of the function** $F(g)$ with increasing disorder for the model of $\epsilon_\beta$ Eq.(90) of Ref. [13] for disorder strengths $W = 3.2 - 20.6$ at the band center $E = 0$. At weak disorder $F(g)$ approaches the parabolic form, while at strong disorder it has a wide segment of quasi-linear behavior. The derivative $\partial_g F(g)$ at $g = 0$ is equal to $1/2$, while the point $g_1 < 0$ where the derivative is 1 is symmetric to the point $g_0 > 0$ of a maximum. The quantity $F(0)$ depends on disorder strength $W$ and decreases with increasing $W$. The localization transition in the infinite system corresponds to $F(0) = -1$.

where $F(g)$ is some independent of $r$ (in the limit of large $r$) function (see Fig. 7) which precise form depends on $\epsilon_\beta$ (for the dependence of $\epsilon_\beta$ on disorder for $K = 2$ Cayley tree see Appendix B) and which is obtained from the following equations:

$$F(g) = \frac{\ln \epsilon_{\beta(g)}}{\ln K} + g\,\beta(g), \tag{73}$$

$$g = -\frac{1}{\ln K}\,\partial_\beta[\ln \epsilon_\beta]|_{\beta=\beta(g)}. \tag{74}$$

The distribution $\mathcal{F}(y = \ln|H_{nm}|)$ for the RP model equivalent to RRG is obtained from Eq. (72) by setting $r = d \approx \ln N / \ln K$:

$$\mathcal{F}(y = \ln|H_{nm}|) \sim N^{F\left(-\frac{2\ln|H_{nm}|}{\ln N}\right)}. \tag{75}$$

Eqs. (70),(71) and (73) define the distribution of $y = \ln|H_{nm}|$ of the form Eq. (3). This is a broad distribution which characteristic width grows with $\ln N$ and its typical value decreases as a negative power of $N$. This form is known as the *large deviation*, or *multifractal ansatz*. It appears in many different problems of statistical mechanics (see e.g. Ref. [66] and references therein) and is a non-trivial generalization of Central Limit Theorem when the logarithm of the fluctuating quantity is a sum of many terms with special correlations between them. An example of such a distribution is shown on the right panel of Fig. 6.

### 7.3 Generic properties of the function $F(g)$ with symmetry Eq.(62).

The symmetry, Eqs. (62),(69) leads to a number of nice properties of the function $F(g)$. From Eqs. (73),(74) and the definition of $g_\beta$ (13) one can immediately obtain:

$$\beta(g_\beta) = \beta. \tag{76}$$

Let us now establish some properties of the function $F(g)$ and some particular values of $g_\beta$ that follow from Eqs. (62),(69). Differentiating Eq. (62) w.r.t. $g$ one obtains:

$$F'(g) + F'(-g) = 1. \tag{77}$$

Substituting $g = g_1$ in this relation we easily obtain:

$$F'(-g_1) = 0.$$ (78)

If the function $F'(g)$ is monotonous and given that $F'(g_0) = 0$ one obtains:

$$g_1 = -g_0.$$ (79)

Now, substituting $g_1$ for $g$ in Eq. (62) and using Eq. (79) we obtain:

$$F(g_1) - g_1 = F(-g_1) = F(g_0) = 0, \Rightarrow F(g_1) = g_1.$$ (80)

Here we also used the normalization condition of the PDF Eqs. (3),(75) that requires:

$$F(g_0) = 0.$$ (81)

Now let us plug $g_{1/2}$ into Eq. (77) and take into account the definition $F'(g_{1/2}) = 1/2$. Then:

$$F'(-g_{1/2}) = F'(g_{1/2}) = \frac{1}{2}.$$ (82)

Now, again, assuming the monotonic behavior of $F'(g)$ one obtains:

$$g_{1/2} = -g_{1/2} = 0.$$ (83)

Next, from Eq. (83),Eq. (73) and Eq. (76) we obtain:

$$F(0) = F(g_{1/2}) = \frac{\ln \epsilon_{\beta(1/2)}}{\ln K} = \frac{\ln \epsilon_{1/2}}{\ln K}.$$ (84)

Another useful relationship one can obtain from Eqs. (74),(13):

$$\partial_g^2 F(g_\beta) = -\frac{\ln K}{\partial_\beta^2 \ln \epsilon_\beta}.$$ (85)

In particular:

$$\begin{aligned}
\partial_g^2 F(0) &= -\frac{\ln K}{\partial_\beta^2 \ln \epsilon_{\beta=1/2}}, \\
\partial_g^2 F(g_0) &= -\frac{\ln K}{\partial_\beta^2 \ln \epsilon_{\beta=0}}.
\end{aligned}$$ (86)

### 7.4 The Lyapunov exponents

In this section we express the fundamental quantities, the Lyapunov exponents, on an infinite Cayley tree in terms of the function $F(g)$ and the eigenvalue $\epsilon_\beta$. The Lyapunov exponents are defined as:

$$\lambda_{typ} = -\lim_{r \to \infty} r^{-1} \langle \ln |G(r)| \rangle,$$ (87)

and

$$\lambda_{av} = -\lim_{r \to \infty} r^{-1} \ln \langle |G(r)| \rangle,$$ (88)

where $G(r) = G_{nj}$ at $|n - j| = r$.

The typical Lyapunov exponent, Eq. (87) is given by the typical value $y_{typ}/r = (g_0/2) \ln K$ of $\mathcal{F}_r(y)$:

$$\lambda_{typ} = \frac{g_0}{2} \ln K = -\frac{g_1}{2} \ln K.$$ (89)

The "average" Lyapunov exponent, Eq. (88) is expressed in terms of $F(0)$ as follows:

$$
\begin{aligned}
\lambda_{av} &= (-F(g_{1/2}) + (1/2) g_{1/2}) \ln K \\
&= -F(0) \ln K = -\ln \epsilon_{1/2}.
\end{aligned}
\tag{90}
$$

To arrive at this result we used the saddle point approximation in computing the average $\langle |G(r)| \rangle$ over the distribution $\mathcal{F}_r(y)$, Eq. (72), and Eqs. (83),(84).

Eq. (90) allows to establish the limits of variation of $F(0)$ in the extended phase on the Cayley tree. Indeed, the localization transition point in an infinite tree is determined by [58]:

$$
\epsilon_\beta = K^{-1}.
\tag{91}
$$

Thus the lower limit of $F(0)$ in the extended phase is $F(0) = -1$. On the other hand, in the case of single orbital per site in the clean limit both Lyapunov exponents tend to $(1/2) \ln K$ (see, e.g. [13]). This corresponds to the upper limit $F(0) = -1/2$.

Note that for a granular tree with $n \to \infty$ orbitals per site (described by a sigma-model) the limit $F(0) = -1/2$ is reached at a finite value of inter-granula conductance. This happens at a point of a transition to an ergodic phase [20] in which $F(0)$ remains equal to $-1/2$. We conclude therefore that for any model on a tree in the extended phase:

$$
-1 > F(0) > -1/2.
\tag{92}
$$

### 7.5 Finite-size multifractality in RRG

Now we are in the position to show the presence of the finite-size multifractality in the MF-RP model with ACTA symmetry (62). Indeed, computing in the saddle-point approximation the averages (9), (10), (11), and (12) with the distribution Eq. (3) and using the properties of the function $F(g)$ established in Sec. 7.3 we obtain the following criteria:

- Localization transition:

$$
1 + F(0) - \ln \left| \frac{4F''(0)}{F''(g_0)} \right| \frac{1}{2 \ln N} + O\left( \frac{1}{\ln^2 N} \right) = 0.
\tag{93}
$$

- Ergodic transition:

$$
1 + F(0) - \frac{\ln \ln N}{2 \ln N} + \ln \left| \frac{2F''(g_0)}{\pi} \right| \frac{1}{2 \ln N} + O\left( \frac{1}{\ln^2 N} \right) = 0.
\tag{94}
$$

- FWE transition:

$$
g_0 = 1.
\tag{95}
$$

One can see from Eqs. (93),(94) that in the thermodynamic limit $N \to \infty$ the criteria for the localization and the ergodic transitions become identical, i.e. the transitions merge together at the tricritical point (see Fig. 1). The absence of a gap between them implies that the multifractal phase collapses in this limit.

We would like to emphasize that this conclusion is heavily based on the ACTA symmetry, Eq. (62), since it is this symmetry that ensures $g_{1/2} = 0$ and $g_1 < 0$. For a distribution, Eq. (3), that does not respect this symmetry (e.g. for the logarithmically-normal one, Eq. (47), with $p \neq 1$) the true multifractal phase does exist if $F(g_{1/2}) > -1$ but $F(0) < -1$ (e.g. for $p < 1$ in Fig. 1).

However, at any finite $\ln N$ the finite-size multifractality appears even if the ACTA symmetry is respected:

$$F(0)|_{W=W_{AT}} - F(0)|_{W=W_{ET}} = -\frac{\ln\ln N}{2\ln N} + \frac{\ln\left|\frac{8F''(0)}{\pi}\right|}{2\ln N}. \qquad (96)$$

Note that the leading double-logarithmic finite-size correction is present for the ergodic transition and is absent for the localization one. This happens because the integral that determines the average in Eq. (9) is dominated by the saddle-point $g = g_{1/2} = 0$, while the saddle point $g = g_1 < 0$ in the average in Eq. (10) is outside the region of integration. This average is dominated by the lower cut-off $g = 0$ where $F'(g) \neq 1$. Therefore, the measure of the *saddle-point* integration $\sim (\ln N)^{-1/2}$ in Eq. (9) is canceled out by the corresponding factor in the normalization constant $C$, Eq. (12), while the measure $\sim (\ln N)^{-1}$ of the *end-point* integration in Eq. (10) is not. One can see from Eq. (96) and Fig. 2 that the leading finite-size correction $(-1/2)\ln\ln N/\ln N$ makes $W_{AT}$ higher than $W_{ET}$. The next correction appears to be also of the same sign at $K = 2$. As the result a finite-size multifractality appears inside a gap between $W_{AT}$ and $W_{ET}$ which is shown in the inset of Fig. 1. At conventional sizes $N = 32000$ accessible for exact diagonalization the finite-size multifractality may extend itself as far as to $W = 14$ down from the infinite-size localization transition $W_c = 18.2$ at $K = 2$ (see Fig. 2).

The emergence of finite-size multifractality for system sizes smaller than the one given by Eq. (96) is of principal importance. For very large $\ln N$ one can expand $F(0) = \ln \epsilon_{1/2}/\ln K$ in $W$ around the infinite-size tri-critical point $W_c$ and define the critical length $L_{MF} = \ln N_{MF}$ which is associated with finite-size multifractality, (8):

$$1 + F(0) \sim \frac{W_c - W}{W_c} \approx \frac{k(\ln\ln N_{MF} + c)}{\ln N_{MF}}. \qquad (97)$$

For $K = 2$ the coefficient $k \approx 0.6$, $c \approx 1.9$.

Note that this length is principally different from the length $L_c \sim \ln N_c \sim (W_c - W)^{-\nu}$ with $\nu = 1/2$ which emerged in the sigma-model description of the granular Cayley tree and RRG in Refs. [54, 55, 67] and in many other works thereafter (e.g. in the recent work [68]). The critical length $L_c$ with the exponent $\nu = 1/2$, is indeed, present in the finite-size scaling (FSS) in localization problem on RRG. It is also present in FSS in an RP model with the ACTA symmetry, e.g. in the logarithmically-normal RP ensemble with $p = 1$ [47]. The critical length $L_{MF}$ that controls finite-size multifractality was earlier suggested [56] but it was believed that it is the same length as $L_c$. However, recently a numerical evidence appeared that the additional critical length with the exponent $\nu = 1$ is also present in problem considered [13, 69, 70].

Eq. (8) that follows from Eq. (97) finally solves this problem and provides a physical meaning for the additional critical length $L_{MF}$ that is much larger than $L_c$.

# 8 Results of exact diagonalization on RRG and on LN-RP ensemble.

In order to confirm the correspondence between RRG and MF-RP developed in the previous section, here we present the results of exact diagonalization for $R_{av}(t)$ both in the logarithmically-normal RP ensemble, Eq. (47) with the ACTA symmetry, $p = 1$, $N = 65536$, and for the Anderson model on RRG with $W = 12$, $N = 131072$, Fig. 8. The similar plots emerge for different values of disorder in the weakly-ergodic phase (see Appendix A). The similarity of the plots for LN-RP ensemble (left-top panel) and for RRG (right-top panel) is obvious. In the inset of both plots we present the log-log derivative that demonstrates a very clear plateau for more than a decade of time which gives the "apparent" stretch exponent $\kappa(N)$. The exponent $\kappa(N)$

depends $N$ and for the system sizes $N = 16384, 32768, 65536, 131072$ it *decreases* with increasing $N$ approximately as $\kappa(N) = \kappa_\infty + a/\ln N$. This extrapolation gives $\kappa_\infty = 0.23$ which is in a reasonably good agreement with the 'theoretical value' $\kappa_{\text{theor}} = 0.193$, see Appendix B. However, the convergence to the thermodynamic limit is almost reached for RRG at $W = 8$ at system sizes $N = 32768, 65536$ (see Fig. 9 and Fig. 10 in Appendix A). In this case the stretch-exponent $\kappa(N = 65536) \approx 0.50$, which should be compared with the theoretical value $\kappa_{\text{theor}} \approx 0.473$ (see Appendix A and B). In Appendix A we also present the results for RRG at $W = 6$ where the convergence to $N \to \infty$ limit is good already at the system size $N = 32768$. In this case we obtain $\kappa(N = 32768) = 0.64$ vs. the theoretical value $\kappa_{\text{theor}} = 0.689$. These results convincingly demonstrate that the mapping of the Anderson model on RRG onto the multifractal RP ensemble is *quantitatively correct*.

## 9 Conclusions and Discussion

In summary, in this paper we introduce a generic set of dense Rosenzweig-Porter random-matrix models with fat-tailed distributions of the off-diagonal elements. The consideration of such models is motivated by the fact that similar models emerge in the mapping of the sparse matrix Hamiltonians of disordered interacting many-body systems [17] and of the Anderson localization model on hierarchical graphs [35] onto the dense random matrix models. The above mapping is characterized by the large deviation ("multifractal") character of the distribution of the off-diagonal matrix elements with the corresponding dependence of the typical value and the width of the distribution on the system size.

We focus on the dynamical properties of such random-matrix models and identify analytically three distinctly different dynamical regimes:

**(i)** Diffusion,

**(ii)** Sub-diffusion, and

**(iii)** "Frozen" dynamics.

Since the above mentioned mapping of models with short-range coupling to the infinite-range RP ones invalidates the notion of distance, in order to describe the dynamics we use, instead of the wave packet spreading, the probability $R(t)$ of a particle initially prepared in a certain graph node (representing a configuration in the Hilbert space) to return back to it.

In the single-particle models on $d$-dimensional lattices, such return probability decays as the inverse (sub)diffusion volume $V(t) \sim [\langle x^2 \rangle]^{d/2} \sim [(Dt)^\kappa]^{d/2}$ occupied by a wave packet at a time $t$. Instead, the Hilbert space of the many-body systems and the Anderson localization model on the hierarchical graphs correspond to the infinite dimensionality and show the exponential growth of the number of nodes with the Hamming distance. This immediately implies the exponential decay of the return probability in the diffusive phase $R(t) \sim e^{-Dt}$. The corresponding sub-diffusive dynamics should then be given by the stretch-exponential decay $R(t) \sim e^{-(Dt)^\kappa}$.

We have shown that the sub-diffusive, stretch-exponential dynamics of $R(t)$ emerges quite generically in the ergodic phase of the "multifractal" RP model in the proximity to the Anderson localization transition. In the toy model of a quantum particle on disordered Random Regular Graph, the sub-diffusive regime spreads throughout the entire ergodic phase down to vanishing disorder. The mechanism of sub-diffusion is related to the rare large couplings which contribution to the decay rate decreases with time. This is essentially a kind of a quantum Griffiths effect which leads to the failure of the Fermi Golden Rule and as such it goes in line with the explanation [36, 45] of slow dynamics in certain quantum many-body systems [71, 72].

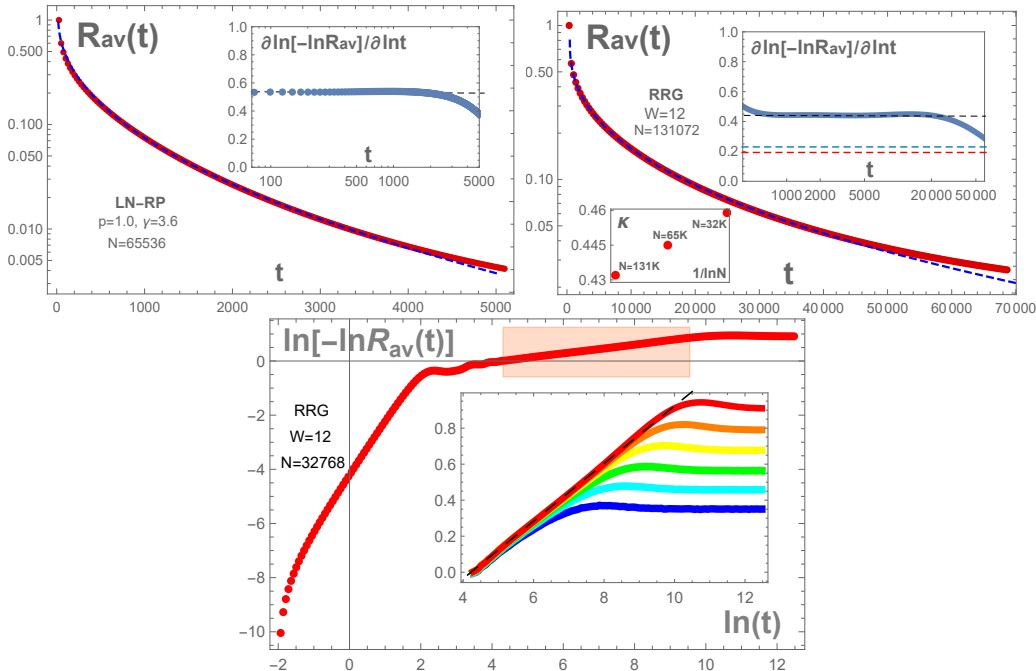

Figure 8: (Color online) **The survival probability in a "multifractal" RP ensemble and in the Anderson model on RRG. (Upper-Left panel)**: The survival probability $R_{av}(t)$ for the logarithmically-normal RP model, Eq. (47) with $p = 1$, in the weakly-ergodic phase, $\gamma = 3.6$ and $N = 65536$, as a function of time. The main plot is obtained by the numerical Fourier transform of the "low-energy" part of the eigenstate overlap correlation function $K(\omega)$ shown in Fig. 9 in Appendix A. In the inset: the plateau in the log-log derivative $d\ln[-\ln R_{av}(t)]/d\ln t$ demonstrates the region of the stretch-exponential behavior. The blue dashed line on the main plot shows the power-law $\ln[R_{av}(t)] \propto -t^{\kappa}$ corresponding to the plateau value of $\kappa(N) \approx 0.53$. **(Upper-Right panel)**: the stretch-exponential behavior of $R_{av}(t)$ for RRG with $W = 12$ and $N = 131072$. For $2000 \lesssim t \lesssim 40000$ it is well approximated by the stretch-exponential function with $\kappa(N) = 0.44$ (blue dashed line on the main plot and the black dashed line in the inset). This value of $\kappa(N)$ suffers from strong finite-size effects and decreases linearly with $1/\ln N$ as $N$ increases (see lower inset). Extrapolation to $N \to \infty$ gives $\kappa = 0.23$ (blue dashed line in the upper inset) which fits reasonably well the "theoretical" value $\kappa_{\text{theor}} = 0.19$ (red dashed line in the upper inset). **(Lower panel)**: the results of direct exact diagonalization numerics for $R_{av}(t)$ for RRG with $W = 12$, $N = 32768$ in the time domain. The region of the stretch-exponential behavior is shown by a pink rectangle. The inset shows the extension of the domain with the stretch-exponential behavior as the system size $N = 1024 - 32768$ (from blue to red) increases.

As a next step in this direction, one can consider the mapping of a true many-body system onto the Rosenzweig-Porter model of the corresponding symmetry, similarly to an oversimplified approach used in Ref. [17] to map several many-body problems onto the Gaussian Rosenzweig-Porter ensemble. This challenging problem consists of two parts. One should first take into account the *essential* graph structure of the Hilbert space of a realistic many-body system, eliminating system-specific details (e.g. the weak links as in the percolation problem [73, 74]). Second, the disordered potential which is uncorrelated on different sites in the coordinate space in the interacting systems, imposes correlations and constraints in the corresponding disorder distribution in the Hilbert space [34]. The effects of the correlated

resonances and the corresponding avalanche structure of delocalization transition in such systems may drastically affect the dynamical phase diagram.

Another relevant direction is to focus on the (sub)diffusive dynamics in the *coordinate* space of the interacting systems instead of the Hilbert space. In order to do this in terms of the local observable like $R(t)$ one should consider the probability of a system to return not to a certain Hilbert space configuration but to a fixed state of a certain local operator (e.g., the occupation number on one site). This fixed state corresponds to an entangled manifold of quantum configurations and should be therefore described by a density matrix. The corresponding generalization of the Wigner-Weisskopf approximation for such a case is the subject of the further investigations.

# Acknowledgements

We are grateful to A. Kudlis and P. A. Nosov for their help in numerical solution of a problem of computing $\epsilon_\beta$ and for insightful discussions.

**Funding information** VEK acknowledges the Google Quantum Research Award "Ergodicity breaking in Quantum Many-Body Systems" and Abdus Salam ICTP for support during this work. IMK acknowledges the support of the Russian Science Foundation (Grant No. 21-12-00409).

# A Numerics on LN-RP model and on Anderson localization model on RRG.

In this Appendix we present the results of the exact diagonalization of 30% to 100% of states in the LN-RP model defined by Eq. (47) and compare these results with the corresponding results for the Anderson model on RRG with $K = 2$.

We start by studying the functions $K(\omega)$ and $\mu(\omega)$, Eqs.(54) and (56), which definitions and principle properties were discussed in Section 6. In Fig. 9 we plot the results of the exact diagonalization for the functions $K(\omega)$ and $\mu(\omega)$ for LN-RP with $p = 1$ respecting the symmetry, Eq. (62) and for RRG with $W = 12$ and $W = 8$ at different system sizes up to $N = 131072$. One can see that there are two different regions of $\omega$: the "low-energy" region, where $K(\omega)$ and $\mu(\omega)$ exhibit similar universal behavior for RRG and the "symmetric" LN-RP model, and the "high-energy" region where the behavior is system-specific and independent of the system size. The energy scale separating the high- and the low-energy parts is the energy scale of establishing of the universal "RP regime". It does not depend on $N$ but depends on disorder strength $W$. In some vague sense it is similar to the inverse mean free path $\ell^{-1}$ in the problem of diffusion in dirty metals. For distances $r < \ell$ the motion of particle is ballistic while for $r > \ell$ a universal diffusive regime is formed. For describing the localization effects one has to take into account interaction between diffusion propagators (of particle-hole and particle-particle type) which is conveniently done in the framework of the nonlinear sigma-model. For such a *coarse-grained* model, the length $\ell$ is the low-distance cutoff. Analogously, the "coarse-graining" in our problem is done by mapping of the full Anderson model on RRG onto an effective Rosenzweig-Porter model.

Let us focus at the "low-energy" part of $K(\omega)$ and $\mu(\omega)$. One can see that at small sizes the curves "move" to the left with increasing the system size $N$ signaling on the $N$-dependence of the characteristic energy scale $E_0 \sim N^{-a}$, $(a > 0)$. In general, such a behavior is a signature of multifractality, including a finite-size multifractality.

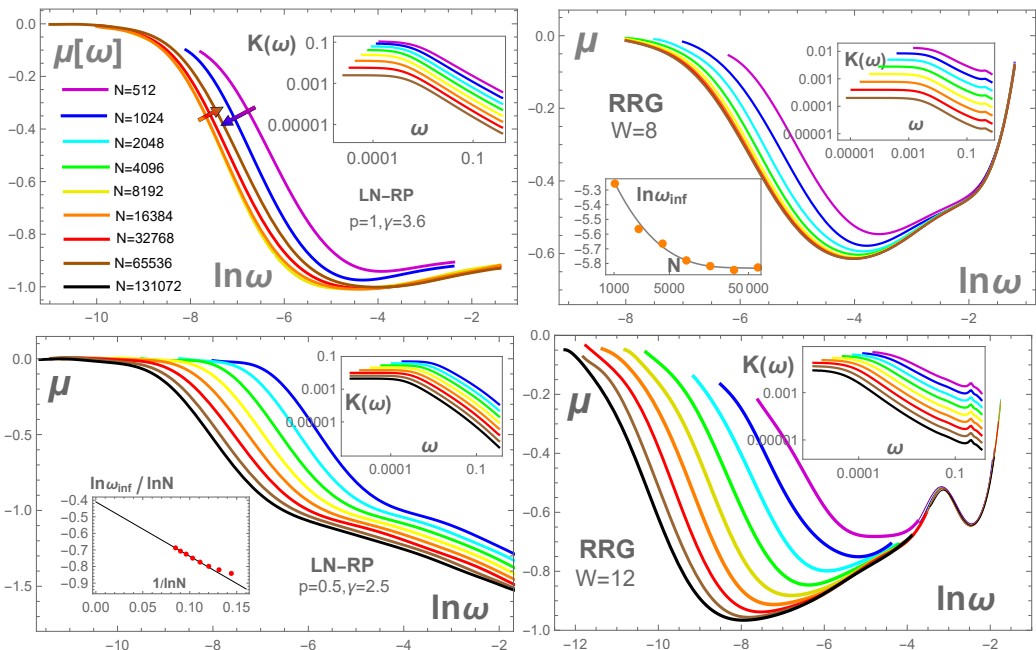

Figure 9: (Color online) **The eigenstate overlap correlation function $K(\omega)$, Eq. (54) and its log-log derivative (the "running power") $\mu(\omega)$, Eq. (56). (Upper-Left panel):** $\mu(\omega)$ for LN-RP model, Eq. (47), with the ACTA symmetry at $p = 1$ and $\gamma = 3.6$ for different matrix sizes $N = 512–65536$. The running power $\mu(\omega)$ interpolates between zero at low frequencies and $\mu \approx -1$ at the minimum. The dependence on the system size shows a re-entrant behavior with the direction of evolutions shown by arrows. It reflects the reduction of the characteristic energy scale $E_0$ at small system sizes due to the finite-size multifractality and the growth of $E_0 \sim E_{BW}$ with $N$ at large system sizes due to $N$-dependence of the total spectral bandwidth $E_{BW}$ (for RRG $E_{BW} \approx W/2$ is size-independent). **(Upper-Right panel):** $\mu(\omega)$ in the Anderson model on RRG with $W = 8$. Lower inset: the evolution of the inflection point on the main plot with increasing the system size. The main plot and this evolution show convergence to the $N = \infty$ limit. The "low-energy" part $\ln \omega < 2$ is similar to that for LN-RP model on the upper-left panel. In the limit $N \to \infty$ we expect both curves to be identical in their low-energy parts if time is measured in units of the corresponding $E_{BW}^{-1}$. **(Lower-Left panel)** $\mu(\omega)$ for LN-RP model ($p = 0.5$, $\gamma = 2.5$) without the ACTA symmetry in its multifractal phase for $N = 1024–131072$. The curve for $\mu(\omega)$ does not show a minimum as a function of $\ln \omega$ but rather a bending point at the level of $-1$. Lower inset: the evolution with $N$ of the inflection point on the main plot. It approaches the power-law $E_0 \sim N^{-0.4}$ in the limit of very large $N$. The characteristic scale $E_0$ decreases in the multifractal phase due to the reduction of the width of a mini-band. **(Lower-Right panel):** $\mu(\omega)$ for RRG at $W = 12$. The "low energy" part $\ln \omega \lesssim -4$ is similar to that in 'symmetric' LN-RP on the upper-left panel, though the convergence to the thermodynamic limit $N \to \infty$ is much slower and does not show re-entrance at system sizes studied. The "high-energy" part at $\ln \omega \gtrsim -4$ is system specific and $N$-independent. The low-energy part $\ln \omega < -4$ was Fourier-transformed to obtain the stretch-exponential behavior on the upper-right panel of Fig. 8. **(Upper insets)** in all panels show $K(\omega)$ for the same parameters as the main panels. The color code (from purple to black) of the curves for system sizes $N = 512, 1024, 2048, 4096, 8192, 16384, 32768, 65536, 131072$ is the same on all the plots and given by the legend in the upper left panel.

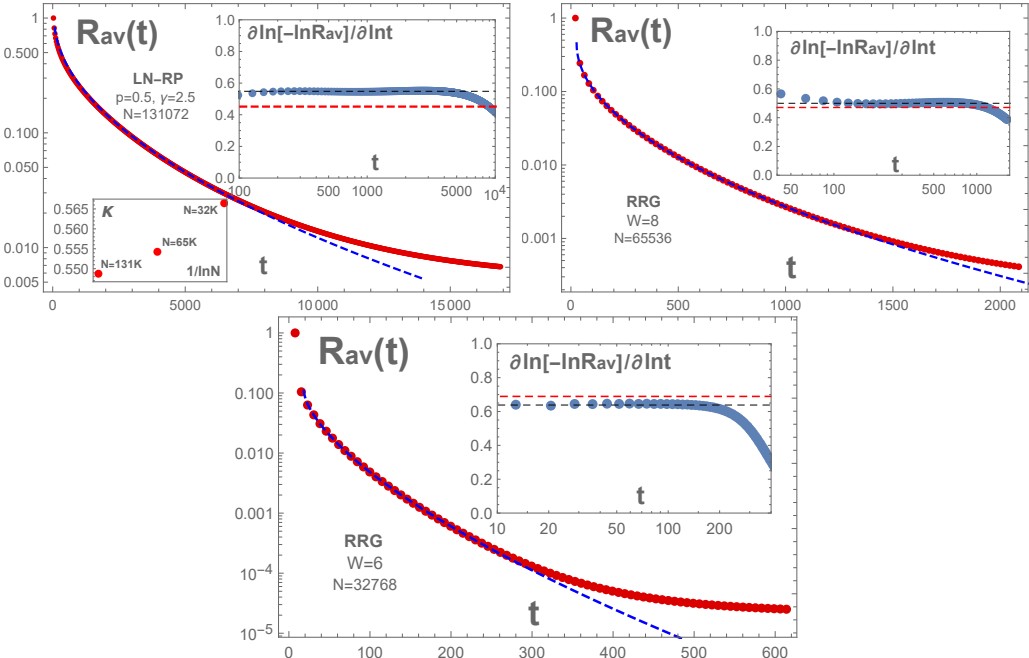

Figure 10: (Color online) **(Upper-Left panel):** The stretch-exponential behavior of the survival probability for LN-RP ensemble, Eq. (47), in the *multifractal* phase ($p = 0.5$, $\gamma = 2.5$ and $N = 131072$). It was obtained by the numerical Fourier-transform of the "low-energy" part $\ln\omega < -1.5$ of $K(\omega)$ in the lower-left panel of Fig. 9. The stretch-exponential behavior $R_{\mathrm{av}}(t) \sim \exp[-(tE_0)^\kappa]$ is qualitatively similar to the one obtained in the *weakly-ergodic* phase of LN-RP model (upper-left panel of Fig. 8), despite the qualitative difference in behavior of $\mu(\omega)$ for $\omega$ above the onset of a shoulder (see left panels of Fig. 9). This demonstrates that only $\omega$ below the onset of the shoulder is relevant for the Fourier transform in the time-interval of the stretch-exponential behavior, where the Levy $\alpha$-stable distribution emerges in $K(\omega)$. **(Upper-Right panel):** The stretch-exponential behavior of the survival probability for the Anderson model on RRG with $W = 8$ and $N = 65536$ in the *weakly-ergodic* phase obtained by the numerical Fourier transform of the "low-energy" part $\ln\omega < -1.2$ of $K(\omega)$ shown on the upper-right panel of Fig. 9. For RRG with $W = 8$ the exponent $\kappa(N = 65536) \approx 0.43 \pm 0.05$ agrees with the value $\kappa_{\mathrm{theor}} = 0.4728$ at $N = \infty$ obtained from Eq. (36) and the exact $F(g)$ (see Appendix B). **(Lower panel):** The stretch-exponential behavior of the survival probability for the Anderson model on RRG with $W = 6$ and $N = 32768$. The exponent $\kappa(N = 32768) = 0.64$ compared to $\kappa_{\mathrm{theor}} = 0.689$.

In the case of a true MF phase in LN-RP model with $p = 0.5$ without the symmetry, Eq. (62), shown on the lower left panel of Fig. 9, the "speed" of this motion approaches a finite limiting value, which is demonstrated in the lower inset of that panel. At large enough $N$ the position of the inflection point $\ln\omega_{inf}$ in $\mu(\omega)$ moves to the left with a "speed" $d\ln\omega_{inf}/d\ln N$ that is extrapolated to $N \to \infty$ linearly in $1/\ln N$ to be $\tau^* = -0.4$. This value is reasonably close to the predicted value $\tau^* = -0.35$ (see Eq.(100) of Appendix C).

However, in the case of the *finite-size* multifractality in the weakly-ergodic phase, as the system size $N$ increases, the "speed" of this "motion" slows down to zero (see upper-right panel of Fig. 9) and may even reverse the sign (upper-left panel of Fig. 9). On RRG for small enough disorder (e.g. for $W = 8$ in upper-right panel of Fig. 9) the "motion" stops at accessible system

sizes showing convergence to some limiting curve in the thermodynamic limit $N \to \infty$. For LN-RP in the weakly-ergodic phase (see upper left panel of Fig. 9) the sign of $a(N)$ changes at some $N$ and becomes positive for large enough $N$. This is in agreement with $E_0 \sim E_{BW} \to N^{-\tau^*}$ with $\tau^* < 0$ (see Eq.(100) in Appendix C) in LN-RP model and $E_0 \sim E_{BW} \sim N^0$ on RRG.

Now let us consider the *shape* of the curves $\mu(\omega)$ in the universal "low-energy" regime. Again, there is a qualitative difference between $\mu(\omega)$ in the weakly-ergodic phase shown in the right and upper panels of Fig. 9 and that in the multifractal phase shown on the lower left panel of that figure. In weakly ergodic phase all the curves for $\mu(\omega)$ on the main plot show a minimum, while in the multifractal phase there is only a bending point on the corresponding level $\mu \approx -1$.

A similar minimum in $\mu(\omega)$ can be extracted from the population dynamics numerics presented in Fig.10 of Ref. [13] (see Fig. 11).

In order to get the correct behavior of survival/return probability $R_{\mathrm{av}}(t)$ we Fourier-transformed $K(\omega)$ *in the entire* "low-energy" region of $\omega$ (see Fig. 9) and checked the results by the direct calculation of $R_{\mathrm{av}}(t)$ in the time domain (see lower panel of Fig. 8). The results are shown in the upper panels of Fig. 8 in the main text and in Fig. 10. All those plots are very similar, no matter in the weakly-ergodic or in the multifractal phase. This implies that in the non-symmetric case of LN-RP when both multifractal and weakly-ergodic phases exist, the stretch-exponential decay of survival/return probability is not sensitive (apart from changing the characteristic scale $E_0$) to the ergodic transition (e.g., when crossing the green line on the phase diagram of Fig. 1 ). This is an argument in favor of the common origin and hidden similarity of both these phases.

## B   Theoretical values of the stretch-exponent $\kappa$ for $K = 2$ RRG.

The "theoretical" value of the stretch-exponent $\kappa_{\mathrm{theor}}$ on RRG can be computed with any prescribed accuracy using the exact theory of localization on a Cayley tree of Ref. [58]. A reformulated variant of this theory and details of calculations of $\epsilon_\beta$ will be published elsewhere [65]. Here we note that in these calculations we solved a non-linear equation for the modified effective distribution of on-site disorder prior to solve the eigenvalue problem for the linearized transfer-matrix equation with this modified distribution. The results for the eigenvalue $\epsilon_\beta$ and the function $F(g)$ in the "multifractal" distribution, Eq. (3), of hopping matrix elements is shown in Fig. 12 for several values of disorder. The "theoretical" value of the stretch-exponent

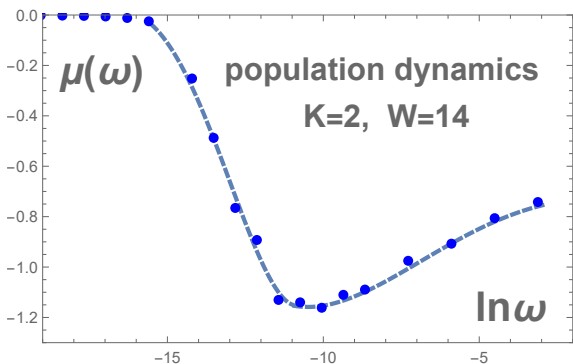

Figure 11: (Color online) $\mu(\omega)$, **defined in Eq. (56)** extracted from the population dynamics (PD) data on $K(\omega)$ presented in Fig.10 of Ref. [13]. The dashed line is merely a guide for eye. The position of the minimum $\mu_{\min} \approx -1.1 < -1$ is likely due to poor statistics.

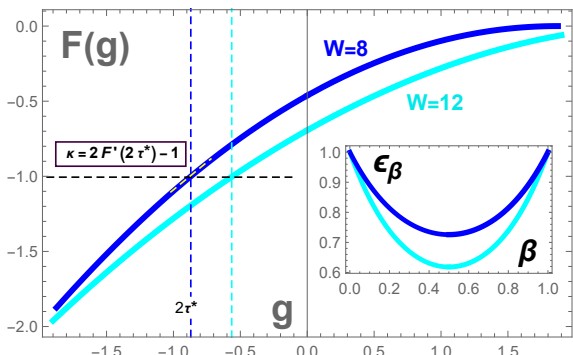

Figure 12: (Color online) **The functions $\epsilon_\beta$ and $F(g)$ and the theoretical value of $\kappa_{\text{theor}}$.** The eigenvalue $\epsilon_\beta$ of the transfer-matrix equation on a tree with the modified on-site energy distribution is found from the numerical solution of a non-linear problem [58, 65] for $K = 2$ Cayley tree for the box-shaped initial disorder distribution with $W = 8$ (dark blue) and $W = 12$ (light blue). The corresponding function $F(g)$ in the distribution, Eq. (3), is computed from Eq. (73). The "theoretical" value of the stretch-exponent $\kappa_{\text{theor}}$ is found from Eqs. (36) and (39) and is determined by the derivative $F'(g)$ at a point where $F(g) = -1$.

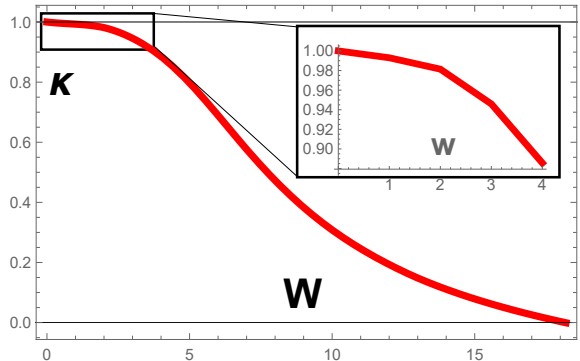

Figure 13: (Color online) $\kappa_{\text{theor}}$ **for RRG with $K = 2$ and box distribution of on-site disorder.**

$\kappa_{\text{theor}}$ is found from Eqs. (36) and (39) and is determined by the derivative $F'(g)$ at a point where $F(g) = -1$, as shown on Fig. 12. We obtained the following values of $\kappa_{\text{theor}}$:

$$\begin{aligned}
\kappa_{\text{theor}}(W = 6) &= 0.6895, \quad \kappa_{\text{theor}}(W = 8) = 0.4728, \\
\kappa_{\text{theor}}(W = 12) &= 0.1933, \quad \kappa_{\text{theor}}(W = 18.17) = 0.0002.
\end{aligned}$$

(98)

The full curve $\kappa_{\text{theor}}$ vs $W$ for *RRG* with $K = 2$ and box distribution of disorder is shown in Fig. 13. The inset on this figure demonstrates that the stretch-exponential phase persists up to vanishing disorder, like the multifractal phase on the finite Cayley tree with one orbital per site [13, 22]. This implies that there is an intimate relation between the two phases, with the SE phase on RRG being a "remnant" of the MF phase on the finite Cayley tree.

## C  Theoretical values of exponents $\kappa$, $\tau^*$ and $\Delta$ for generic LN-RP model

Finally we present expressions for the stretch-exponent $\kappa$, Eq.(36), the exponent $\tau^*$ of the characteristic scale $E_0 \sim N^{-\tau^*}$, Eq.(37), and for the exponent $1-\Delta$ of the total spectral band-width $E_{BW} = N^{1-\Delta}$, Eq.(38), for the generic LN-RP model.

$$\kappa = \begin{cases} 1, & \gamma < \min\left(2, \frac{1}{p}\right) & [E] \\ \frac{2}{\sqrt{p\gamma}} - 1, & \frac{1}{p} < \gamma < \min\left(4p, \frac{4}{p}\right) & [WE-SE] \\ \sqrt{\frac{4-(2-p)\gamma}{p\gamma}}, & \max(2, 4p) < \gamma < \frac{4}{2-p} \text{ and } p < 1 & [MF-SE] \\ 0, & \gamma > \min\left(\frac{4}{2-p}, \frac{4}{p}\right) & [WE-F] \end{cases}, \tag{99}$$

$$-\tau^* = \begin{cases} \frac{1-\gamma(1-p)}{2} > 0, & [WE-E, FE-E] \\ \sqrt{p\gamma} - \frac{\gamma}{2} > 0, & [WE-SE, WE-F] \\ 1 - \gamma(1-p) < 0, & [MF-E] \\ -\frac{\gamma(1-p) - \sqrt{(\gamma(1-p))^2 + \gamma(4p-\gamma)}}{2} < 0, & [MF-SE] \end{cases}, \tag{100}$$

$$1 - \Delta = \begin{cases} \frac{1-\gamma(1-p)}{2} > 0, & [WE-E, FE-E] \\ \sqrt{p\gamma} - \frac{\gamma}{2} > 0, & [WE-SE, WE-F] \\ 0, & [MF] \end{cases}. \tag{101}$$

As it follows from Eq.(100) the characteristic energy scale $E_0 \sim N^{-\tau^*}$ decreases with increasing $N$ in the multifractal phases (in this case its physical meaning is the width of a mini-band in the local spectrum) and increases with $N$ in the ergodic phases (in this case it is of the order of the total spectral bend-width $E_{BW}$). The band-width $E_{BW} \sim N^{1-\Delta}$ is blowing up with increasing $N$ in the ergodic phases, while it is constant in the non-ergodic ones (see Eq.(101)).

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
