# Peer review of "Dynamical phases in a ``multifractal'' Rosenzweig-Porter model"

_SciPost Physics, doi:SciPost Phys. 11, 045 (2021)_

## Round 1 · Referee Report · Anonymous (Referee 1) · 2021-7-4

Report

The authors convincingly argue in the introduction that the Porter-Rozenzweig type random matrices with heavy-tail (''multifractal'' or ''large deviation type'') distributed off-diagonal entries may serve as a valid model for grasping essential features of multifractal eigenstates in various regimes/phases typical for MBL systems. The most essential new result is a careful dynamical analysis of survival probability showing that in a broad region of parameters the stretch-exponential behavior of such object is generic, implying subdiffusion. Moreover, the authors show that there is a genuine (in the thermodynamic limit) phase transition between phases with exponential and stretch-exponential decay. In fact dynamical approach reveals new phases which can not be detected by simply looking at eigenfunctions in static approach. Thus dynamical phases reflect effects different from those detected by ergodicity violation criteria. The analysis is based on carefully explored Wigner-Weiskopff approximation which allows to take into account the finite-size effects which turn out to be of crucial importance for correctly interpreting the earlier numerical results and resolving the long-standing controversies in a convincing way. A crucial feature not much discussed before is the revealed existence of a tricritical point. This point seems to be very essential for reliable interpretation of numerics as in its vicinity a new, parametrically different correlation length arises not seen in previous analysis.

In summary, this is a high quality paper of broad interest, well written and nontrivially contributing to a topic of active research interest. I suggest it is published after the authors consider the minor remarks below.

1) page 13, after eq.(30): 'and unity with a polynomial correction' Sounds cryptic for me, please reformulate

2) Is (33) a definition of the exponent \Delta?

3) Eq.(34) seems to be the definition of \tau_*, which should be clearly stated (mentioned in words in fig 4 but better to repeat it in the text)

4) when discussing the analysis of the Fourier transform (52)-(55) for this class of models I believe it is appropriate to mention the paper Y.V. Fyodorov and A.D. Mirlin, Phys. Rev. B, vol. 55, R16001 (1997) where this type of correlator was addressed for sparse random matrix ensemble closely related to RRG.

---

## Round 1 · Referee Report · Anonymous (Referee 2) · 2021-7-26

Strengths

This paper provides a rather detailed treatment of the localization properties of a class of random matrix models. More specifically, the models analysed are Rosenzweig Porter models in which the off-diagonal matrix elements have a broad distribution. The phase diagram for the models is established, both according to static criteria (are eigenstates localized, multifractal, weakly or fully ergodic?) and according to dynamic criteria (how does a wavepacket - initially concentrated on one basis state - spread in time?).

The strengths of the paper are:

  1. Explicit and complete results for the Rosenzweig-Porter model with a log-normal distribution of off-diagonal matrix elements, including some finite-size effects.
  2. Clear presentation of the background for the work, including an approximate mapping to the RP model from the problem of Anderson localization on random regular graphs.
  3. Clear presentation of the essentials of calculations.

Weaknesses

The main weakness of the paper, as I see it, is that the problem treated is a very specialized one.

Report

While the problem treated is, as I have noted, a specialized one, the results are important in relation to an extensive body of past work. This work is concerned with viewing many-body localization in Fock space, making links to Anderson localization on the random regular graph, and with localization of the random regular graph. On those grounds, the paper can be argued to meet the acceptance criteria by "Opening a new pathway in an existing research direction ...".

In my opinion the general acceptance criteria are also met.

---

## Editorial Decision

published